# Molecular and Supramolecular Structure of the Mitochondrial Oxidative Phosphorylation System: Implications for Pathology

**DOI:** 10.3390/life11030242

**Published:** 2021-03-15

**Authors:** Salvatore Nesci, Fabiana Trombetti, Alessandra Pagliarani, Vittoria Ventrella, Cristina Algieri, Gaia Tioli, Giorgio Lenaz

**Affiliations:** 1Department of Veterinary Medical Sciences, Alma Mater Studiorum University of Bologna, 40064 Ozzano Emilia, Italy; fabiana.trombetti@unibo.it (F.T.); vittoria.ventrella@unibo.it (V.V.); cristina.algieri2@unibo.it (C.A.); 2Department of Biomedical and Neuromotor Sciences, Alma Mater Studiorum University of Bologna, 40138 Bologna, Italy; gaia.tioli2@unibo.it

**Keywords:** oxidative phosphorylation, respiratory supercomplexes, ROS, ATP synthase/hydrolase, mitochondrial dysfunction, mitochondrial permeability transition pore, *cristae*, cellular signaling

## Abstract

Under aerobic conditions, mitochondrial oxidative phosphorylation (OXPHOS) converts the energy released by nutrient oxidation into ATP, the currency of living organisms. The whole biochemical machinery is hosted by the inner mitochondrial membrane (mtIM) where the protonmotive force built by respiratory complexes, dynamically assembled as super-complexes, allows the F_1_F_O_-ATP synthase to make ATP from ADP + Pi. Recently mitochondria emerged not only as cell powerhouses, but also as signaling hubs by way of reactive oxygen species (ROS) production. However, when ROS removal systems and/or OXPHOS constituents are defective, the physiological ROS generation can cause ROS imbalance and oxidative stress, which in turn damages cell components. Moreover, the morphology of mitochondria rules cell fate and the formation of the mitochondrial permeability transition pore in the mtIM, which, most likely with the F_1_F_O_-ATP synthase contribution, permeabilizes mitochondria and leads to cell death. As the multiple mitochondrial functions are mutually interconnected, changes in protein composition by mutations or in supercomplex assembly and/or in membrane structures often generate a dysfunctional cascade and lead to life-incompatible diseases or severe syndromes. The known structural/functional changes in mitochondrial proteins and structures, which impact mitochondrial bioenergetics because of an impaired or defective energy transduction system, here reviewed, constitute the main biochemical damage in a variety of genetic and age-related diseases.

## 1. Introduction: Functions of Mitochondria. The Oxidative Phosphorylation System

The advancement of molecular medicine has pinpointed the role of mitochondria in the etiology and pathogenesis of most common chronic diseases [1,2,3,4], so much that the term “Mitochondrial Medicine” has been proposed [5] and then widely used [6,7,8,9].

The early biochemical studies on mitochondria were centered on their role in energy conservation. The energy-transducing membrane-bound enzyme complexes of inner mitochondrial membrane (mtIM) drive biochemical reactions involved in energy transformation or bioenergetics. Oxidation of substrate by respiratory complex and ATP production by ATP synthase are tightly coupled molecular mechanisms in the oxidative phosphorylation (OXPHOS) system as explained by Mitchell’s chemiosmotic hypothesis [10].

Once the major aspects of the OXPHOS system were clarified, the interest in mitochondria somewhat decreased. However, in recent years it has been raised again due to a series of novel findings assigning new roles to mitochondria in molecular and cell biology, such as mitochondrial DNA and mitochondrial genetics, the role of mitochondria in generation of reactive oxygen species (ROS) and in cell signaling, in cellular quality control and apoptosis (programmed cell death). Nevertheless, these newly discovered functions are strictly intertwined with the central role of electron transfer and ATP synthesis.

This review intends to outline the major structural and functional aspects of mitochondrial bioenergetics that are at the basis of changes leading to pathology; in particular, we will deal with recent advances of the supramolecular structure of the respiratory chain complexes and of F_1_F_O_-ATP synthase/hydrolase; for this reason, this review will not analyze other important aspects in much detail such as the intimate mechanisms of electron transfer and proton translocation on one hand, and the analysis of the individual pathologies on the other.

## 2. The Respiratory Chain of Mitochondria

The major mechanism of energy conservation in eukaryotes is OXPHOS, performed by a multi-enzyme system embedded in the mtIM and constituted by two portions: the respiratory chain and the ATP synthase complex.

The membrane electron transfer system (membrane-ETS) is a series of enzymes that collects electrons which stem from the oxidations of intermediary metabolism and drives them downhill to oxygen molecules which are reduced. The free energy fall that accompanies electron flux creates an electrochemical proton gradient (Δμ_H+_) since H^+^ are pumped from the mitochondrial the matrix, namely the compartment inside the mitochondrion, to the intermembrane space (IMS) localized between the inner and outer mitochondrial membranes [11]. The energy associated to the proton gradient is then largely used to synthesize ATP from ADP and Pi by the ATP synthase complex. The ATP synthesized is transferred to the cytoplasm in exchange with ADP by the ATP/ADP translocase, also exploiting the H^+^ gradient.

As previously revised in [12], it was Hatefi et al. [13] who first isolated from mitochondria four enzyme multi-subunit complexes that concur on the oxidation of NADH and succinate, namely NADH-Coenzyme Q reductase (Complex I, CI), succinate-Coenzyme Q reductase (Complex II, CII), ubiquinol-cytochrome *c* reductase (Complex III, CIII or cytochrome *bc*_1_ Complex) and cytochrome *c* oxidase (Complex IV, CIV) [14].

The connection among these enzyme complexes is ensured by two mobile transporters of electrons, i.e., Coenzyme Q (CoQ, ubiquinone) and cytochrome *c* (cyt. *c*) [14]. The former is a lipophilic quinone incorporated in the lipid bilayer of the mtIM, while cyt. *c* is a hydrophilic hemoprotein facing the mitochondrial IMS, in contact with the external surface of the mtIM. The membrane-ETS operates through the following sequence of the respiratory enzyme complexes: (Equations (1) and (2))
NADH → CI → CoQ → CIII → cyt. *c* → CIV → O_2_(1)
or
succinate → CII → CoQ → CIII → cyt. *c* → CIV → O_2_(2)

In addition, the membrane-ETS consists of other proteins having electron transfer activity [12] that converge at the CoQ junction. Glycerol-3-phosphate dehydrogenase is involved in a shuttle of reducing equivalents from cytosol to mitochondria [15], electron transfer flavoprotein (ETF) dehydrogenase is involved in fatty acid oxidation [16], dihydroorotate dehydrogenase catalyzes a step of pyrimidine nucleotide biosynthesis [17], choline dehydrogenase is important for the regulation of phospholipid metabolism [18], and sulfide dehydrogenase is involved in the disposal of sulfide [19]. A schematic representation of mammalian respiratory chain is found in Figure 1.

For reviews see, e.g., Sousa et al. [20] and, for individual complexes: Parey et al. [21] for CI, Bezawork-Geleta et al. [22] for CII, Xia et al. [23] for CIII, and Zong et al. [24] for CIV.

### 2.1. Organization of the Respiratory Chain: Historical Outline

The advancement of science is long and tortuous: it often happens that novel discoveries, rather than providing further clarifying information on the existing basic knowledge, bring doubts and contradictory issues. This happened for the discovery of the respiratory chain supercomplexes (SCs) and for the elucidation of their function. Therefore, we perform a concise historical survey of the respiratory chain, since some early findings came out before the present knowledge, but their significance was not understood.

As we mentioned in the previous section, Hatefi et al. [13] accomplished the systematic resolution and reconstitution of four functional respiratory complexes from mitochondria, and proposed that the overall electron transfer from substrates to oxygen results from both intra-complex and inter-complex redox reactions: intra-complex electron transfer takes place in the “solid” state of redox components (e.g., flavins, FeS clusters, cytochromes) having fixed steric relation, whereas inter-complex electron transfer operates by rapid diffusion of the mobile components acting as co-substrates, i.e., CoQ and cyt. *c* [14]. In the following years, this proposal was confirmed by the kinetic analysis of Kröger and Klingenberg [25,26], leading Hackenbrock et al. [27] to postulate the *Random Collision Model of Electron Transfer*:

“Electron transport is a diffusion-coupled kinetic process; Electron transport is a multicollisional, obstructed, long-range diffusional process; The rates of diffusion of the redox components have a direct influence on the overall kinetic process of electron transport and can be rate limiting, as in diffusion control… It is concluded that mitochondrial electron transport is a diffusion-based random collision process, and that diffusion has an integral and controlling effect on electron transport.”

The random collision model was accepted by most researchers in this field. Nevertheless, the accumulation of further experimental evidence obtained with newly developed techniques has led to the proposal of a different model of supramolecular organization based upon specific interactions between individual respiratory complexes [28].

Indeed, a supramolecular assembly of respiratory components had been already present in the pioneering studies of Chance and Williams [29], who described the respiratory chain as an aggregate of coenzymes (flavins and cytochromes) in solid-state, embedded in a protein matrix.

Surprisingly, evidence for a non-random arrangement of respiratory complexes also derived from the same early investigations of Hatefi’s group, reporting isolation of CI+CIII units [30], suggesting that such units preferentially assemble in the native membrane. These aggregates were considered physiological by their discoverers, indicating the existence of supramolecular units of electron transfer. It is therefore clear that SCs, as well as the entire NADH oxidase (known as the respirasome, see later) had been isolated and reconstituted. The publication of the fluid mosaic model of biomembranes [31] was applied also to mitochondrial membranes, thus favoring the random collision model of Hackenbrock. Thus, Hatefi’s idea of specific associations among complexes was overlooked.

The idea of fixed associations, however, was never completely abandoned: in fact, some authors reported the possible existence of specific associations between respiratory complexes, either fixed [32] or dynamic [33].

At the turning of the century, Schägger and Pfeiffer [34] applied the technique of Blue-Native Gel Electrophoresis (BN-PAGE) to digitonin-solubilized yeast and mammalian mitochondria and found electrophoretic bands of high molecular weight indicating specific associations among respiratory complexes. According to their discoverers, these associations represent the state of the respiratory chain under physiological conditions. The same study also provided evidence for a dimeric ATP synthase complex.

The supramolecular arrangement of the respiratory complexes is well established [35,36], although the functional role of respiratory SCs is still controversial and their relationship with a random distribution of the individual complexes partially remains to be clarified [37,38,39,40].

### 2.2. Distribution and Composition of Respiratory Supercomplexes

All SCs exhibit highly ordered structures, thus they are not the result of artificial protein–protein interaction due to the extraction and solubilization procedures [41].

In most studies, SCs appear to contain the three “core” respiratory complexes, i.e., CI, CIII, and CIV, whereas the other respiratory enzymes may be randomly distributed in the lipid bilayer [12]. These “core” complexes have the common feature of carrying out proton translocation and of having subunits encoded by mitochondrial DNA.

Respiratory SCs have been described and characterized in the mitochondria of several mammalian tissues and in many other organisms, also including plants, fungi and bacteria [42].

The supramolecular structural organization of the “core” respiratory complexes CI, CIII, and CIV is conserved in all higher eukaryotes. The SC I_1_III_2_IV_1–4_ contains all three complexes required for the complete oxidation of NADH by molecular oxygen, and for this reason it was called “respirasome”. SCs containing only two components are also normally found in large amounts, as the SC I_1_III_2_ in which CIV is not present, as well as SCs III_2_IV. In bovine heart mitochondria, only a small aliquot of CI is present in free form in the presence of digitonin [43]; thus, it is believed that all CI is bound to CIII in the native membrane.

A higher level of complexity in the organization state of SCs may be due to the presence of additional protein components unrelated to electron transfer complexes (e.g., Shy1, SCAF1, Rcf2, HIGD2A) occasionally found by BN-PAGE [44].

Wang et al. [45] described a multifunctional mitochondrial fatty acid β-oxidation (FAO) complex that is physically associated with SCs. Analysis of genetic defects in both fatty acid oxidation and OXPHOS [46] revealed that some patients with primary FAO deficiencies exhibit secondary OXPHOS defects. These metabolic interrelations support the view that OXPHOS proteins and FAO are physically associated, and that these interactions are critical for both functions.

The other respiratory enzymes not comprising the “core” of the proton translocation machinery appear not to be associated in SCs [47]. In most studies of BN-PAGE, CII was not found associated with other complexes of the respiratory chain. In addition, mitochondrial glycerol phosphate dehydrogenase is absent in the respirasome.

Fang et al. [48] observed that mitochondrial function is hampered by dihydroorotate dehydrogenase deficiency; the enzyme was shown to physically interact with respiratory complexes CII and CIII by immunoprecipitation and BN/SDS/PAGE analysis.

Sulfide-quinone oxidoreductase and sulfite oxidase, involved in sulfide oxidation, which transfer electrons to the respiratory chain, were also described to be associated with SCs containing CIV [49].

## 3. Supercomplexes May Provide a Kinetic Advantage to Electron Transfer

The discovery of SCs led to the proposal that these aggregates form to improve substrate channeling or enhance catalysis in inter-complex electron transfer. Substrate channeling is the direct transfer of an intermediate between the active sites of two enzymes which catalyzes reactions which occur one after the other [50]; in the respiratory chain, this means that electrons are directly transferred between two consecutive enzymes by alternate reduction and re-oxidation of an intermediate which is not diffused in the medium. In such a case, inter-complex electron transfer cannot be distinguished from intra-complex electron transfer. Therefore, the mobile intermediates predicted to exhibit substrate-like behavior in the random collision model, i.e., CoQ and cyt. *c*, would be buried in the interface between two consecutive complexes within the SC.

In the CoQ region, the occurrence of electron transfer by channeling of CoQ between adjacent CI and CIII contrasts with the so-called CoQ “pool behavior”, previously sustained by the kinetic analysis of Kröger and Klingenberg [25]. To better understand the following sections, we believe it necessary to briefly mention their kinetic analysis. These authors showed that steady-state respiration can be described as a simple two-enzyme system, where the first enzyme reduces ubiquinone (V_red_) while the second one oxidizes ubiquinol (V_ox_), kinetically behaving as a homogeneous pool. Thus, the total CoQ molecules must randomly link any number of the dehydrogenase(s) with any number of the oxidase(s); the overall steady-state activity (V_obs_) is related to V_red_ and V_ox_ as shown by the so-called pool equation: (Equation (3))
Vobs = (Vred × Vox)/(Vred + Vox)(3)

It has been shown that cyt. *c* also exhibits “pool behavior ” [51].

In the following sections, the relations existing between the two models and the ensuing controversial findings will be analyzed. The main challenge to study the physiological role of SCs is to selectively disrupt respiratory SCs in the cell [40].

### 3.1. Structural Evidence Suggesting Channeling

#### 3.1.1. Molecular Structure of Supercomplexes

If channeling occurs between CI and CIII by the common intermediate CoQ, the redox groups involved in CoQ reduction by CI and CoQH_2_ re-oxidation in CIII must be in close contact in order to form a driving pathway containing CoQ itself; similar reasoning applies to cyt. *c* between CIII and CIV. An essential requirement for this condition is the availability of a high-resolution map of the molecular structure of the SCs.

In the initial studies, purified SCs were analyzed by negative-stain electron microscopy and single-particle cryogenic electron microscopy (cryo-EM) [52,53]. Pseudo-atomic models of the respirasome (SC I_1_III_2_IV_1_) were produced by fitting the known X-ray structures of the component complexes to the 3D maps of the respirasome. Evidence for channeling in the respirasome stems from the observed distinct arrangement of the three component complexes, in which the CoQ-binding sites in CI and in CIII face each other and are divided by a gap within the SC membrane core, which most likely contains lipids. Althoff et al. [52] proposed that CoQ may cover a trajectory through such a 13 nm-long gap.

Further high-resolution studies by cryo-EM of mammalian mitochondrial SCs [54,55,56,57] may be useful to predict the CoQ pathway between CI and CIII, as shown in Figure 2.

**Figure 2 life-11-00242-f002:**
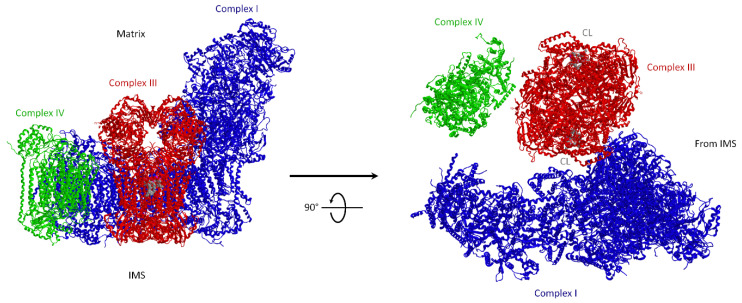
Bovine mitochondrial supercomplex (SC) I_1_III_2_IV_1_. Fitted model by single particle cryo-EM. Side view (on the left) and view from the IMS (on the right) showing two CL (cardiolipin) molecules (in space filling mode) in the cavity of each monomer of CIII formed by cytochromes *c*_1_ and *b*. Image modified from Mileykovskaya and W. Dowhan [58]. The enzyme subunits are drawn as ribbon representations obtained from modified PDB ID code: 2YBB. Blue, CI; red, CIII dimer; green, CIV.

In theory, channeling within the SC may occur through either close docking of the active sites with real tunneling of substrate among proteins, or covering relatively long distances presumably by substrate-restricted diffusion (microdiffusion) within the space between the two active sites; both alternatives share a channeling which necessarily occurs between two fixed sites of the same SC. Microdiffusion is kinetically distinguished from bulk diffusion (pool behavior), where the interaction of the substrate molecules stochastically occurs by collisional encounters with a large number of possible sites in distinct respiratory complexes attained by random diffusion.

Information on channeling within the SC molecular structure must be compatible with the known CoQ reduction mechanisms by CI and its re-oxidation by CIII dimers. Ubiquinol oxidation occurs via a cyclic mechanism known as the Q-cycle [59,60]. On considering the functional asymmetry of the CIII dimer [61], it is suggestive to consider that in the supercomplexed CIII dimer only one monomer is active for NADH-dependent respiration and CoQ-channeling, since the CoQ reducing site I CI and the Qo site where ubiquinol is reoxidized are close to each other. On the contrary, the other monomer, which lacks such constrictions, might be easily available to interact with the CoQ pool at the (distal) Q_i_ site.

The group of Kuhlbrandt [57] showed that the CoQ reduction site of CI in the SC is 11 nm far from the proximal CIII monomer, whereas the distal one is 18 nm away, thus suggesting that the latter is functionally inactive. The linear CI arrangement with the active CIII monomer and with CIV is strongly in favor of channeling for electron transfer between complexes in the respirasome by substrate microdiffusion.

On the other hand, the respirasome structure reported by Letts et al. [54] indicates that both CoQ interaction sites in CI and CIII are separated and easily accessible to the membrane, and are likely to provide no limit to free CoQ diffusion in the inner mitochondrial membrnae (IMM).

The SC structure, resolved at up to 3.8 Å in four distinct states, suggests that CoQ oxidation may be rate limiting because of unequal access of CoQ to the active sites of CIII_2_ [62]. CI changes between “closed” and “open” conformations, while a key transmembrane helix rotates in a striking way. Moreover, the CI state influences the conformational flexibility within CIII_2_, demonstrating crosstalk between the enzymes.

According to Letts et al. [62], under high turnover conditions, individual CIs may overcome the capacity for CoQ pool balance, so that the local CoQH_2_ pool may attain higher concentration than in the bulk membrane. By ensuring close association of CI and CIII_2_ in SCs, but still allowing free CoQ exchange with the bulk, the SCs would prevent any local accumulation of CoQH_2_, without limiting CI’s activity by making it entirely dependent on the adjacent CIII_2_ for the CoQH_2_ re-oxidation.

The structural analysis of SCs allows analogous considerations for cyt. *c* between CIII and CIV.

The structural evidence of purified SCs by cryo-EM adds structural reasons for the looseness of CIII–CIV interactions in mammalian mitochondria. In ovine mitochondria, Letts et al. [54] identified two distinct arrangements of SC I_1_III_2_IV_1_: a major “tight” form and a minor “loose” form. In both structures, the density for CIV is weaker relative to that for CI and CIII, indicating the greater conformational flexibility of CIV. In the tight respirasome, CIV contacts both CI and CIII, whereas CIV in the loose respirasome has defined contacts only with CI. Tight and loose architectures may represent independent structures or may interconvert each other, according to distinct stages of assembly or disassembly. The presence of two respirasome forms with distinct CIV linkages was also detected by Sousa et al. [57] and ascribed to instability of the purified SC.

#### 3.1.2. Supercomplexes Are Dynamic Structures: The Plasticity Model

Together with kinetic evidence by flux control analysis in favor of CoQ channeling (See Section 3.2.2), we also postulated [63] that the SC I_1_III_2_ may be in equilibrium with free CI and CIII. In fact, we found that NADH-cyt. *c* reductase activity in reconstituted proteoliposomes can occur either by channeling or by random diffusion, depending on the protein density in the lipid bilayer of the proteoliposomes.

Acín-Pérez et al. [64] in a study of purified SCs showed by BN-PAGE that other types of associations exist besides the respirasome, such as CI+CIII or CIII+CIV. They therefore suggested that a variety of associations between respiratory complexes is likely co-exists in vivo with free complexes and proposed an integrated model, the plasticity model, for the organization of the respiratory chain. According to their view, the previous contrasting models, solid vs. fluid, are only the two extremes of a dynamic range of molecular associations between respiratory complexes [64]. The plasticity model proposes that both CoQ and cyt. *c* are sequestered into distinct sub-domains and are both trapped into SCs and freely diffusible. This promiscuity can stem either from the dissociation of SCs or from the escape of CoQ or cyt. *c* from the supramolecular assemblies [36,65].

The plasticity model and the dynamics and stability of mitochondrial SCs [37] are still debated issues. The putative factors involved in association/dissociation of SCs, such as membrane potential and post-translational changes, are reviewed by Genova and Lenaz [42]. However, even if indirect evidence shoulders the SC dynamic assembly and disassembly, up to now no direct demonstration exists that SCs physiologically form and dissociate in vivo.

#### 3.1.3. The Role of Lipids: Cardiolipin in Supercomplexes

SCs have 2–5 nm gaps at the transmembrane interfaces of the individual complexes (Figure 2) that are presumably filled with lipids. Wu et al. [56], in their high-resolution structure of the purified SC I_1_III_2_IV_1_ from porcine heart, localized many phospholipid molecules including cardiolipin (CL) involved in the protein–protein interactions [66]. Substantial progress was made by studying Barth syndrome, a genetic syndrome characterized by severe cardiopathy, in which cardiolipin remodeling is altered due to the mutation of the gene *Tafazzin*; in Barth syndrome patients SCs are unstable, leading to a functional impairment characterizing the disease [67]. In immortalized lymphoblasts from Barth’s syndrome patients the amount of SCs is decreased, as well as the amount of individual CI and CIV [68].

Direct involvement of CL in the formation of SC III–IV was demonstrated in genetically manipulated strains of *S. cerevisiae* in which the CL content can be regulated in vivo. Yeast mutants lacking CL have normal amounts of individual complexes, but lack the stable SC III_2_IV_2_ as observed in the wild type parental strain [69]. Since *S. cerevisiae* lacks CI, these studies could not provide information on SCs containing CI.

In contrast with CL depletion that destabilizes SCs, the loss of phosphatidylethanolamine (PE) favors the formation of larger SCs between CIII and CIV in *S. cerevisiae* mitochondria. The reason why CL and PE, even if both associate in non-bilayers, act in an opposite way on SC stability may depend on the different charge, since, at physiological pHs CL is anionic and PE is zwitterionic. Using yeast mutants of PE and phosphatidylcholine (PC) biosynthesis, Baker et al. [70] showed a specific requirement for mitochondrial PE, but not PC, in CIII and CIV activities, but not for their formation. Neither PE nor PC were involved in respiratory SC formation, emphasizing the specific requirement of CL in SC assembly.

Our laboratory has stressed the importance of CL in stabilizing CI-containing SCs. Reconstitution of binary CI/CIII proteoliposomes from bovine heart mitochondria at high lipid to protein ratio (30:1 w:w) prevents formation of SC I_1_III_2_ [71]. However, SC I_1_III_2_ and related NADH-cyt. *c* reductase activity are maintained if these high-lipid proteoliposomes are enriched with 20% CL (w:w), resembling the percent content of CL in the mitochondrial membrane (M. Kopuz, Y. Birinci, S. Nesci, G. Lenaz and M.L. Genova, unpublished data). Likewise, the dilution of native protein-bound CL in the excess exogenous phospholipids prevents SC assembly, but the effect is reversed by increasing the CL content of the liposomes, therefore shifting again the equilibrium to CL binding to protein.

Addition of reactive oxygen species (ROS) to mitochondria affects the respiratory activity due to cardiolipin peroxidation, since CL is required for the optimal activity of respiratory complexes [72,73]. We showed by flux control analysis (cf. Section 3.2.2) that the SC I_1_III_2_ in proteoliposomes disappears if lipid vesicles are peroxidized before protein reconstitution [71]. Evidently, the distortion of the lipid bilayer induced by peroxidation and the alteration of the phospholipid annulus originally present in the purified SC I_1_III_2_ determines its dissociation.

CL deficiency was reported to cause a defect in respiratory function and a decrease in ATP synthesis [74], an indirect demonstration of CL role in the structure of the SCs. Rieger et al. [75] investigated the effect of ALCAT1 overexpression. This enzyme catalyzes the incorporation of polyunsaturated fatty acids into CL. The resulting CL species are more susceptible to oxidative damage, leading to increased ROS production and mitochondrial dysfunction. Using galactose as sugar supply, cells must employ OXPHOS for ATP synthesis, shown as a significant increase in basal- and ADP-linked respiration; ALCAT1 overexpression prevents the galactose-induced increase in respiration.

Likewise, a genetic defect of CL synthase induces loss of SC assembly and of respiratory activity as well as other mitochondrial defects [76].

### 3.2. Kinetic Evidence for Channeling in the Coenzyme Q Region

#### 3.2.1. Rate Advantage Imposed by SCs

The overall rate in a diffusion-coupled pathway is always less than that of the rate-limiting step and only approaches the latter when it is widely different (i.e., much slower) compared to the rate of the other step(s) in the pathway [25]. On the contrary, the rate is expected to be equal to the rate of the limiting step if channeling of the intermediate substrate(s) occurs.

Ragan and Heron [77] investigated in detail the reconstitution of CI in the respiratory chain; they showed that purified CI and CIII, when mixed as concentrated solutions, reversibly associate in a 1:1 molar ratio to form an SC I+III endowed with NADH-cyt. *c* reductase activity. The study demonstrated a stoichiometric behavior for this activity, revealing the existence of an active SC formed by CI and CIII: electron transfer is fast within the SC and conversely very slow from the SC to free CIII via the CoQ pool.

On the other hand, prevailing CoQ-pool behavior could be induced and CI and CIII could be made to operate independently of each other if the SCs were functionally dissociated into “free” complexes, by raising the amount of phospholipid and ubiquinone in the concentrated mixture. Under this condition, electron transfer from CI to CIII is still ensured by CI+CIII_2_ units that however dissociate and reform at rates exceeding the rates of electron transfer in the individual complexes.

According to Heron et al. [78], the mobility of CI and CIII in the membrane is lost and the complexes are frozen in their SC assembly when phospholipids are not in excess with respect to the content needed to form a lipid annulus around the protein. Such frozen state favors a stable orientation of the site of CoQ reduction by CI with respect to the site of oxidation by CIII. They also showed that protein-bound CoQ_10_ leaks out of the SC I+III in the lipid bilayer when an extra phospholipid is present in the proteoliposomes; the consequent decrease in activity could be reversed by adding additional quinone. As a corollary, we may deduce that a large amount of ubiquinone found in the natural membrane is needed to maintain the CoQ_10_ content in the SC unit when it is formed.

Our laboratory provided a direct evidence that the phospholipid amount affects the choice of channeling with respect to CoQ-pool behavior [79]. A proteoliposome system obtained by fusing a crude mitochondrial fraction enriched in CI and CIII with different amounts of phospholipids and CoQ_10_ allowed us to discriminate between these two mechanisms. The experimental NADH-cyt. *c* reductase activity was compared with the theoretical values obtained from the pool equation of Kroger and Klingenberg [25], showing overlapping results in the range from 1:10 to 1:40 (w/w) protein to lipid ratios. However, pool behavior was not shown at low (1:1 w/w) lipid to protein ratio. Moreover, the observed rates of NADH-cyt. *c* reductase were higher than the theoretical values; this 1:1 ratio corresponds to the mean nearest distance between respiratory complexes in mitochondria.

Several studies point out that SC disorganization by several causes is accompanied by decreased electron transfer and energetic efficiency (cf. also Section 3.1.3).

Recently Garcia-Poyatos et al. [80] in Enriquez’s group investigated the physiological role of SCs, generating two null allele zebrafish lines for supercomplex assembly factor 1 (SCAF1). The *scaf1^−/−^* fish showed altered OXPHOS activity due to the disrupted interaction of C III and CIV.

Solsona-Vilarrasa et al. [81] studied the effects of increased cholesterol levels, as occurring in vivo in alcoholic and non-alcoholic fatty liver, or by in vitro cholesterol enrichment of mouse liver mitochondria. Cholesterol feeding caused oxidative stress and mitochondrial GSH (mGSH) depletion, which lead to liver steatosis and damage. The overload of cholesterol in mitochondria disrupts mitochondrial functional performance and the organization of respiratory SC assembly, thus potentially contributing to oxidative stress and liver injury.

Tomkova et al. [82] showed that Tamoxifen-resistant cells show a significant decrease in mitochondrial respiration accompanied by a decrease in mitochondrial respiratory SC and significantly increased levels of mitochondrial superoxide (See Section 5).

Balsa et al. [83] showed that endoplasmic reticulum (ER) stress and glucose deprivation stimulate mitochondrial OXPHOS and formation of respiratory SCs by action of protein kinase R-like ER kinase (PERK). Accordingly, PERK genetic ablation or pharmacological inhibition abolishes nutrient and ER stress-mediated increase in SC levels and reduce OXPHOS-dependent ATP synthesis.

These studies suggest that electron transfer between CI and CIII in NAD-linked respiration can take place either by CoQ channeling within the SC I_1_III_2_ or by a less efficient random diffusion behavior, depending on the state of the membrane lipids: nevertheless, it appears that channeling is preferred under physiological conditions.

#### 3.2.2. Metabolic Flux Control Analysis

Our laboratory first exploited flux control analysis [84] to establish the functional association between individual respiratory complexes in beef heart mitochondria [63]. Using specific inhibitors to establish the extent of metabolic control exerted by each individual complex over the entire NADH-dependent respiration, we demonstrated that CI and CIII are both rate-limiting, since they exhibit flux control coefficients (FCC) close to 1, thus behaving as a single enzymatic unit (e.g., SC I_1_III_2_); we therefore concluded that electron transfer through CoQ is accomplished by channeling between CI and CIII [63]. Using the same method for succinate oxidation we found that CII, but not CIII, is rate-limiting, confirming the notion that CII does not form SCs and that the oxidation of succinate follows pool behavior.

A decade later, Blaza et al. [85], on the basis of flux control analysis, criticized the evidence for channeling in the respiratory chain: accordingly, rotenone competes with CoQ and, consequently the CI inhibition extent is affected by the additional quinone analog used as substrate in the CI assay, but absent in the NADH aerobic oxidation assay. Indeed, using rotenone and CoQ_1_ as substrate they find an FCC for CI exceeding 1 that is certainly artefactual. However, in our study we used decyl-ubiquinone (DB) rather than CoQ_1_. DB has much lower affinity than CoQ_1_ for CI [86], thus exerting lower competition with rotenone and lower influence on the inhibition efficiency measured by us [63]. Moreover, the FCC values found by Blaza et al. [85] using alternative inhibitors of CI (e.g., piericidin and diphenyleneiodonium, not competitive with CoQ) are unrealistically low, indicating the presence of a rate-limiting step downhill. Most likely, the limiting amount of cyt. *c* in their sample would shift the main control of the chain to the cyt. *c* region. Significantly, in Blaza’s study, addition of exogenous cyt. *c* to the mitochondrial samples enhanced the FCC from 0.19 to 0.67.

Overall, we must emphasize that our major clue [36] to demonstrate association of CI to CIII was not so much the high FCC of CI but the high FCC of CIII in NADH oxidation, which is incompatible with a CoQ-pool model where CI and CIII independently float in the membrane. Flux control of CIII was not performed in the study by Blaza et al. [85]. Such high FCC value of CIII over NADH-dependent respiration was measured by inhibitor titration with mucidin, and is clearly not artefactual because the corresponding FCC of CIII measured by using the same inhibitor in succinate oxidation is low (as expected from the rate-limiting role of CII and the lack of SC II+III). It is evident that the FCC of CIII is close to 1, indicating a rate-limiting step in CIII, only if the preceding COQ-reductase is assembled with CIII in an SC, which is the case only for CI, but not for CII. In addition, the high FCC that CIII shows over NADH-cyt. *c* reductase activity in reconstituted proteoliposomes is strongly lowered when the SC I_1_III_2_ is dissociated [71] by reconstitution in excess phospholipids or in peroxidized phospholipids.

The few other studies addressed to SCs using metabolic control analysis have confirmed the possibility that the respiratory chain is arranged as functionally relevant supramolecular structures [87,88,89].

#### 3.2.3. Coenzyme Q Compartmentalization

The possibility that the CoQ pool may not be homogeneous has been advanced long time ago. Gutman and Silman [90] observed partial additivity and competition between succinate oxidation and NADH oxidation. On the basis of their observations (and those of others) [91], they proposed a compartmentalization of the CoQ pool, in which two sub-domains partially interact through a spill-over diffusion-mediated mechanism.

Gutman [92] also investigated NADH and succinate oxidation in submitochondrial particles (SMP), as well as the rates of energy-dependent reverse electron transfer from succinate to NAD^+^ and of forward electron transfer from NADH to fumarate, concluding that:

“The electron flux from succinate dehydrogenase to oxygen (forward electron transfer towards Complex III) or to NADH dehydrogenase (reverse electron transfer) employs the same carrier and is controlled by the same reaction” whereas “the electron transfer from NADH to oxygen does not share the same pathway through which electrons flow in the NADH-fumarate reductase”.

In other words, CI and CII (reverse electron transfer) are linked by a different pathway with respect to CI and CIII (forward electron transfer), even if CoQ is a common intermediate for both pathways. We can clearly interpret this non-homogeneity of the CoQ pool with respect to succinate and NADH oxidation as deriving from sequestration of CoQ in the SC I+III in contrast with the free CoQ molecules that connect CII and CIII.

In past years, several other reports have suggested that the CoQ pool is not homogeneous and questioned its universal validity [93]. Similar observations were made by other authors [94] who observed different functional CoQ pools in mitochondria.

Lapuente-Brun et al. [95] demonstrated that the physical association of CI and CIII determines a preferential pathway for electrons mediated by a dedicated subset of CoQ molecules. This compartmentalization would prevent significant cross talk and mixing up between NADH oxidation (CI-dependent) and succinate oxidation (dependent on CII) or other flavoenzyme-dependent oxidations. Moreover, the CIII units within the SC I+III are exclusively dedicated to oxidation of NADH whereas free CIII is dedicated to oxidation of other substrates using the free CoQ pool (e.g., fatty acids through ETF, succinate, glycerol-3-phosphate and choline).

On the contrary, Blaza et al. [85] showed that the steady-state rates of aerobic NADH and succinate oxidation were only partly additive in bovine heart submitochondrial particles SMP; moreover, cytochromes *b*_H_, *b*_L_, *c* and *c*_1_ in the same cyanide-inhibited particles were reduced to a similar extent by either NADH or succinate or a mixture of the two substrates. Blaza et al. [85] interpreted the results as a demonstration that a single homogeneous pool of CoQ molecules exists that receives electrons indifferently from CI and CII. Note that these data were obtained on the whole pathway of electrons from ubiquinol to oxygen and therefore also comprise the steps through cyt. *c* and CIV. It is therefore not possible to discriminate whether the pool behavior is due to the homogeneity of the CoQ-pool or of the cyt. *c* pool.

To overcome this uncertainty, we carried out experiments on succinate and NADH oxidation by exogenous cyt. *c* in KCN-inhibited mitochondria, thus shortening the electron route for both oxidation reactions by only including CoQ and CIII as redox partners [36]. Under such conditions, NADH and succinate oxidation by cyt. *c* was completely or almost completely additive and close to the theoretical sum of the two individual reactions, thus suggesting that two separate CoQ-compartments may exist.

Notably, we also obtained similar results (Tioli G, Falasca AI, Lenaz G and Genova ML, data communicated at EBEC2016-Session P4) using proteoliposomes enriched in CI, CII, and CIII. In these proteoliposomes, the additivity of NADH and succinate oxidation decreased at increasing CIII inhibition by mucidin, thus substantiating the hypothesis that the CoQ molecules in the SC I+III are in a dissociation equilibrium with the molecules in the pool [96], cf. next section.

Fedor and Hirst [97] in another study incorporated an alternative quinol oxidase (AOX) into mammalian heart mitochondrial membranes to have another competing pathway for ubiquinol oxidation. Since AOX strongly increased the rate of aerobic NADH oxidation, they concluded that the quinol generated in SCs by CI is more rapidly reoxidized in the CoQ pool by AOX than by CIII inside the SC. Based on these results, Hirst [98] concludes that quinone and quinol diffuse freely in and out of SC: no substrate channeling occurs, since it is not required to support respiration.

In our opinion, this criticism is inconsistent for the same reasons discussed above for concomitant NADH and succinate oxidation in presence of a CIII inhibitor, since Fedor and Hirst used KCN to block electron transfer through CIV: under this condition electrons from ubiquinol are forced to follow the alternative pathway by dissociation into the CoQ pool (cf. Section 3.3.1). It is intuitive that channeling is largely predominant during high turnover of the respiratory chain, whereas at low turnover, CoQ diffusion in the pool would preferentially take place.

Recently, our analysis was strongly supported by Szibor et al. [99] who expressed AOX in mouse mitochondria and observed that electron input by succinate (CII), but not by NADH (CI), almost completely reduces the Q pool (>90%) irrespective of the respiratory state. AOX enhances the forward electron flux from CII and decreases reverse electron transport. Interestingly, AOX does not act on CI substrates, except in the presence of a respiratory inhibitor (as proven by Fedor and Hirst [97]).

Recently, Enriquez’s group [100] also presented results confirming that the SC of CI and CIII allows the partial segregation the CoQ pool and allows substrate channeling. Their results with different cellular models suggest that when CI is not superassembled, as in CIII-KO cells, CoQ occurs in a unique pool, whereas CI superassembly triggers the formation of two partially different CoQ pools.

### 3.3. If Electron Transfer Occurs via Channeling, What Is the Role of Coenzyme Q Pool?

Since a mobile pool of CoQ in the mtIM coexists with CoQ sequestered in SCs, we may ask whether this pool is a mere reservoir of an excess of CoQ molecules without a specific function or whether the CoQ pool is in any way required for physiological electron transfer and/or for additional functions.

#### 3.3.1. Dissociation Equilibrium of Bound Coenzyme Q

Compelling evidence that electron transfer from CI to CIII occurs physiologically by channeling/restricted diffusion stems from the observation that CI is almost totally associated in an SC with CIII. However, we reasoned that an excess of entrapped CoQ in the pool is also required for channeling to occur [96]. In fact, the CoQ molecules bound in the SC, that ensure electron transfer directly from CI to CIII are in dissociation equilibrium with the CoQ pool. Therefore, the extent of CoQ binding to the SC depends on the equilibrium with free CoQ in the pool and on the size of the pool itself. The existence of this equilibrium is widely confirmed by a series of studies, such as the previously reported findings by Heron et al. [77] on CI to CIII association, by the saturation kinetics for CoQ exhibited by the integrated activity of CI and CIII (NADH-cyt. *c* oxidoreductase) [101] and by the decrease in respiration in mitochondria fused with phospholipids causing subsequent dilution of the CoQ pool [102].

In conclusion, besides acting as a diffusible substrate of several flavin reductases not involved in SC organization (cf. next section), free CoQ molecules are a reservoir for binding to the SC. Studies on respiration under pathological conditions [103,104] showed that SC assembly is required for proper respiration, even if activity of the individual complexes is normal. On the other hand, reconstitution in vitro as described in the previous Section 3.2, suggests that respiration can occur in both modes (channeling and diffusion), if CoQ is available. In a proteoliposome system where CI was reconstituted together with an alternative oxidase (AOX) and CoQ_10_, and in absence of SC, Jones et al. [105] found high rates of NADH oxidation through AOX, demonstrating that CI can deliver electrons through the CoQ pool.

The dynamic character of CoQ bound within the SC, which is in dissociation equilibrium with the free pool of CoQ in the membrane, is the major point favoring controversy concerning channeling. We propose that the CoQ molecules trapped in a lipid micro-domain within the SC and are channeled from CI to CIII during electron transfer at steady state, without significant exchange with the free pool; however, CoQ dissociation from the SC to the pool becomes significant when electron transfer in the respiratory chain is slow or blocked by an inhibitor. In this respect, the plasticity model would be a functional rather than structural feature of the respiratory chain, a possibility taken into account by Enriquez himself [37]. According to this idea, the SCs are stable structures that do not readily dissociate under physiological conditions, while CoQ is said to behave in a highly dynamic fashion: CoQ channeling and diffusion shall occur in proportions depending on the turnover rates of the respiratory chain.

#### 3.3.2. Electron Transfer between Free Complexes

Several flavoenzymes reduce CoQ as outlined in Section 2. Among these, the most widely investigated is CII. All evidence converges in the statement that electron transfer from CII to CIII takes place only through the CoQ pool. Accordingly, succinate oxidation kinetically follows pool behavior after extraction and reconstitution [101] and in intact mitochondria [106] in accordance with the notion that CII does not participate in SC formation (see previous sections). For the same reason, also energy-dependent reverse electron transfer from succinate to NAD^+^, taking place through sequential activity of CII and CI connected by CoQ, must take place by collisional interactions in the CoQ pool. The hyperbolic relation experimentally found by Gutman [92] between the rate of reverse electron transfer and succinate oxidase is in complete accordance with the pool equation.

Other CoQ reductases such as glycerol-3-phosphate dehydrogenase, ETF dehydrogenase, dihydroorotate dehydrogenase, choline dehydrogenase, sulfide dehydrogenase, that are present in minor amounts, are probably inserted in the respiratory chain by interaction through the CoQ pool [12], but kinetic evidence is poor.

Some knowledge exists for mitochondrial glycerol phosphate dehydrogenase (mtGPDH). A study [107] demonstrated that in brown adipose tissue (BAT) mitochondria the inhibition curve of glycerol phosphate-cyt. *c* reductase is sigmoidal in the presence of myxothiazol or antimycin A. Such type of inhibition suggests the occurrence of a homogeneous CoQ pool between mtGPDH and CIII [26]. CIII reduction by mtGPDH in human neutrophil mitochondria was also shown to occur in absence of SC organization and of NAD-linked respiration [103], providing further evidence that mtGPDH functions through the CoQ pool. Preliminary studies by BN-PAGE (M.L. Genova, M. and H. Rauchova, unpublished) showed that mtGPDH is not apparently linked to any of the respiratory complexes. Mráček et al. [108] demonstrated that mtGPDH associates into homo-oligomers but also in high molecular weight SCs of more than 1000 kDa whose composition is unknown, but not associated with CI, III, or CIV as shown by BN-PAGE analysis. No association was found between CI or other OXPHOS complexes and the electron transferring flavoprotein (ETF) that participates in fatty acid oxidation.

Indirect evidence, however, suggests that fatty acyl CoA oxidation proceeds through a separate CoQ pool with respect to NADH oxidation (cf. Section 3.5).

### 3.4. Electron Transfer through Cytochrome c: Is There Evidence for Channeling?

BN-PAGE clearly shows that a fraction of CIV, as well as CIII, participates in SC assembly (cf. Section 2). However, metabolic flux control analysis [63] showed that CIII has a high FCC, as well as CI, in NAD-linked respiration, whereas CIV has a low FCC; this means that CI and CIII behave as a functional enzyme unit, whereas CIV seems to be functionally independent.

These results cannot be easily explained [109]. It is true that most of CIV appears to be free in the BN-gels (cf. Section 2), and one might consider that the cytochrome *c* oxidase activity in the respirasome is masked by the large excess of active enzyme randomly distributed in the membrane. On the other hand, this would mean that channeling is not a major feature of electron transfer in the cyt. *c* region. The latter means that the free molecules of CIV are involved in electron transfer from NADH and implies that the molecules of CIV assembled in the respirasomes are not involved in the channeling of cyt. *c* [38,42], but do behave similarly to the free CIV units in electron transfer by random diffusion.

The conclusion that the absence of channeling in the cyt. *c* region of mammalian mitochondria is a physiological feature, is supported by the observation that the respirasome of potato tuber mitochondria is completely functional in cyt. *c* channeling, as detected by the same flux control analysis in mammalian mitochondria [71]. This difference may be ascribed to a tighter binding of cyt. *c* in the potato respirasome, as shown by BN-PAGE.

The structural evidence of purified SCs by cryo-EM adds structural reasons on the looseness of CIII–CIV interactions in mammalian mitochondria. In ovine mitochondria, Letts et al. [54] identified two distinct arrangements of SC I_1_III_2_IV_1_: a major “tight” form and a minor “loose” form (resolved at the resolution of 5.8 Å and 6.7 Å, respectively). In both respirasome structures, CIV has a weaker density than the other complexes in the respirasome, indicating the greater conformational flexibility of CIV. In the tight respirasome, CIV contacts both CI and CIII, whereas CIV in the loose respirasome only interacts with CI. Tight and loose architectures may represent independent structural entities or may interconvert, indicating different stages of assembly or disassembly. The presence of two forms of the respirasome having distinct CIV linkages was also detected by Sousa et al. [57] and ascribed to instability of the purified SC.

In partial contrast with the above considerations, Lapuente-Brun et al. [95] demonstrated that at least part of CIV forms a functional SC with channeling of cyt. *c*, but the SC formation depends on the availability of the SC assembly factor SCAF1. They showed that when CIV participates in SCs due to the presence of SCAF1, a significant proportion of CIV activity is not utilized in cell respiratory activity. They also identified three CIV populations, one of which is dedicated exclusively to receive electrons from NADH oxidation (forming SC I+III+IV), presumably by cyt. *c*. channeling. The second population receives electrons from FAD-dependent enzymes (forming SC III+IV and presumably operating by cyt. *c* channeling), whereas the third major one is in free form and receives electrons from both NADH and FADH_2_ oxidization, presumably using the free cyt. *c* pool. On the contrary, if CIV is maintained permanently detached from SCs by elimination of SCAF1, all CIV is in free form and electron transfer takes place via diffusion of a single pool of cyt. *c*.

Functional evidence for cyt. *c* channeling was also found in *S. cerevisiae* [110] mitochondria which are characterized by having all CIV bound to CIII in a supercomplex [111], thus preventing electron transfer through free CIV units. The structural evidence by single particle cryo-EM sustains channeling, since in the III_2_IV_2_ SC (there is no CI in this yeast species) the distance between the binding sites of cyt. *c*, i.e., cytochrome *c*_1_ of CIII and the Cu_A_-subunit II of CIV, is considerably shorter than that in bovine mitochondria [112]. At odds with this structural evidence, however, Trouillard et al. [113] showed that the time-resolved oxidation of cyt. *c* by CIV in yeast mitochondria is a random-collision process.

Rydström Lundin et al. [114] studied the kinetics of aerobic quinol oxidation in *S. cerevisiae* mitochondria as a function of the respiratory SC factors Rcf1 and Rcf2 that mediate supramolecular interactions between CIII and CIV forming SCs III_2_IV_1-2_. They demonstrated that Rcf1 promotes formation of a direct electron-transfer pathway from CIII to CIV via a tightly bound cyt. *c*. Accordingly, in these mitochondria under steady-state conditions in the presence of added homologous cyt. *c*, the direct electron transfer through the bound cyt. *c* (i.e., cyt. *c* channeling), is faster than the equilibration of electrons with the cyt. *c* pool. Interestingly, when using heterologous cyt. *c* (from horse heart), electron transfer between CIII and CIV occurs only via the cyt. *c* pool. Realistic theoretical assumptions indicate that electron transfer between CIII and CIV can become rate limiting. Hence, there is a kinetic advantage of bringing CIII and CIV together in the membrane to form SCs [115]. In a yeast mutant defective in SC formation, but containing fully functional individual complexes, the diffusion of cyt. *c* between the separated complexes is delayed, thus reducing electron transfer efficiency [116].

Moreover, other results indicate that in the absence of Rcf1, the interactions required for stability of the SC III+IV are disrupted [117]. Consistently, Rydström Lundin et al. [114] demonstrated that direct electron transfer does not take place in the absence of Rcf1 and electrons are only transferred via the cyt. *c* pool in the Rcf1Δ strain.

In the next Section 3.5, we will discuss the possible physiological reasons why there is no major channeling of cyt. *c* in mammalian mitochondria, contrary to yeast or plant mitochondria.

### 3.5. Supercomplexes and Regulation of Metabolic Fluxes

Does CoQ channeling occur under physiological conditions? As discussed in the previous Section 3.3, the presence of a bottleneck downhill CoQ might induce interaction of the NADH pathway (via channeled CoQ in SC I_1_III_2_IV_n_) with the succinate pathway (CoQ pool behavior) by shifting the CoQ dissociation equilibrium. However, if a rate-limiting step is situated upstream, i.e., in or before the dehydrogenases, the reducing pressure of CoQ on its partner oxidases (particularly in the case of CIII assembled in the SC I_1_III_2_IV_n_) may not be present and the two routes would take place independently. On the other hand, in mammalian mitochondria, it appears that the two fluxes mix up in the cyt. *c* region where most evidence excludes cyt. *c* channeling (Section 3.4).

The existence of two different functional CoQ compartments, one for NADH oxidation an another one for oxidation of succinate and other FAD-linked substrates, has deep implications for the metabolic adaptation to the feeding state. The flux of electrons in the respiratory chain is conditioned by the orientation of metabolism to use different fuels. The use of different fuel molecules generates different proportions of NADH and FADH_2_, that require an optimal equilibrium between the corresponding routes of electron transfer in the respiratory chain. In a well-fed individual, the oxidative metabolism largely follows the glycolytic pathway and the Krebs cycle, so that the oxidation of NAD-linked substrates exceeds succinate oxidation (ca. 5:1) and the electron flux proceeds mainly through SC I_1_III_2_. On the other hand, metabolic adaptation of liver mitochondria to fasting forces fat mobilization and FAO [118]. In fact, an increase in the FADH_2_-dependent respiration, as in FAO, induces saturation of the CoQ pool reoxidation capacity and promotes reverse electron transport from ubiquinol to CI [119]. The resulting local generation of superoxide triggers protein degradation of CI subunits by oxidative damage, and consequent disintegration of the complex, as well as lipid peroxidation, disassembly of SCs and further instability of their enzyme components. Thus, we can infer that CoQ redox status acts as a metabolic sensor that fine-tunes the configuration and supramolecular organization of the respiratory chain in order to match the prevailing substrate profile [120]. In order to avoid such saturation of the CoQ pool oxidation capacity, fasting conditions require proper disassembly of SCs to achieve adjustment of respiratory SC proportions favoring the FAD-linked route. Similar adaptations may occur when the increase in FAO results in response to high-fat diet. Impairment of this adaptation may be relevant to pathological processes associated to obesity.

As postulated by Lapuente-Brun et al. [95], at the metabolic level, the fast kinetic adjustments of the mitochondrial respiration are followed by gene expression changes in the level of individual respiratory complexes.

Why appreciable channeling may not exist at the level of cyt. *c* in mammalian mitochondria? (cf. Section 3.4). Mammalian mitochondria have several dehydrogenases directing electrons to CoQ, however they usually have only one oxidase (CIV) receiving electrons from CIII via cyt. *c*. For this reason, it may be worth speculating that, whilst it may be useful to separate the major NADH-dependent flux from CI from those departing from succinate, fatty acid oxidation and other metabolic pathways by separating the CoQ compartments, there is no such need for cyt. *c* that is by and large receiving electrons univocally from CIII. This assumption is reinforced by the fact that, contrary to what is found in mammalian mitochondria, CIII and CIV of plant mitochondria are functionally operating as an SC and, at the same time, cyt. *c* is tightly bound in the SC, as revealed by flux control analysis [63] and BN-PAGE [71]. Not surprisingly, plant mitochondria are characterized by a high branching of the electron transfer pathways feeding electrons directly to cyt. *c* [121], which requires adjustment of the different routes as a response to physiological needs, as it happens in mammalian mitochondria at the CoQ level. In plant mitochondria, segmentation might be achieved by regulating different compartments of free and bound cyt. *c*.

## 4. Overview of the F_1_F_O_-ATP Synthase/Hydrolase and Its Supramolecular Structure

The mitochondrial respiratory chain and its arrangement in SCs, detailed in the previous sections, has the main bioenergetic role to build the electrochemical gradient which allows the F_1_F_O_-ATP synthase to build ATP, namely, to accomplish the so-called energy transduction, which converts a transmembrane proton movement into chemical energy, which can be used for all the energy-consuming processes of the cell.

The rotary ATPase family, which embraces structurally similar membrane-bound enzyme complexes working as energy transduction mechanisms, consists of three sub-families, A-, V- and F-type ATPases, which originate from a common evolutionary ancestor. A-type ATPases occur in Archaea and some bacteria, V-type ATPases are typical of eukaryotic vacuoles, while F-type ATPases occur in eukaryotic mitochondria, tylakoid membranes of chloroplasts and in bacterial cell membranes [122]. In mitochondria, the hydrophilic (F_1_) and hydrophobic (F_O_) ATPase domains are joined laterally by a single stalk stator and in the middle by a central stalk. The F-type ATPases can function as ATP synthesis or ion pumps coupled to ATP hydrolysis, a bifunctional mechanism unique in nature [123,124]. The F_1_F_O_-ATPase has a mushroom shape, in which F_O_ is membrane-embedded and F_1_ protrudes outside the mtIM. The enzyme complex is universally known as the nano-machine that produces ATP, the “molecular energy currency” under aerobic conditions [125,126]. The hydrophilic enzyme domain basically consists of a spherical extrinsic hexamer formed by three catalytic β subunits alternated with three non-catalytic α subunits. The (αβ)_3_ hexamer contains in the core the asymmetrical central stalk (in turn composed by γ, δ, and ε subunits). The central stalk is joined to the loop of each *c* subunit hairpin. The number of *c* subunits is species dependent: these subunits are arranged as a cylindric palisade to form the *c*-ring. This subunit assembly, namely the *c*-ring and the central stalk, makes up the rotor. The membrane-embedded *a* subunit with unusual “horizontal” hairpin helices named H5–H6 matches the concave barrel-like *c*-ring where the H^+^ sites lie and perfectly fits the ring size in order to create the H^+^ translocation pathway. A static structure peripheral to the rotor, which spans for the entire enzyme complex length, acts as a stator to prevent the (αβ)_3_ rotation torque of the central stalk. The peripheral stalk is composed by various subunits, namely the hydrophilic oligomycin sensitivity-conferring protein (OSCP), F6, *b*, *d*, linked to F_1_-catalytic domain and embedded in the mtIM by the hydrophobic portion of *b* and A6L subunits. The peripheral stalk is also associated to the supernumerary membrane subunits (*sms*) *e*, *f*, *g*, and 6.8-kDa proteolipid (6.8PL). However, 6.8PL was erroneously believed to be inserted in the central hole of the *c*-ring [127] and another supernumerary subunit, diabetes-associated protein in insulin-sensitive tissue (DAPIT), in the tetrameric F_1_F_O_-ATPase porcine model was misassigned, since recently it was reported as localized at the furthest edge of F_O_ domain [128] (Figure 3). So, the emerging advances in the enzyme knowledge make researchers continuously re-consider and re-evaluate the F_1_F_O_-ATPase structure and function.

In mammalian mitochondria, this astonishing enzyme complex has recently revealed versatile roles, in addressing cells to life or death, in the maintenance of the mitochondrial morphology and raised great expectations as a drug target [129,130]. Accordingly, the modulation of the enzyme functions, often compromised by mitochondrial dysfunctions associated with severe diseases, may contribute to innovative therapeutic strategies at the molecular level.

### 4.1. The F_1_F_O_-ATP Synthase from the Energy Production to Mitochondrial Morphology

In the energy-transducing mtIM, the synthesis of ATP is based on the coupling of the rotary mechanisms of the two main sectors, namely the hydrophobic F_O_ and the hydrophilic F_1_ [131,132]. The protonmotive force (Δ*p*), built by substrate oxidation in the respiratory chain, drives H^+^ translocation through F_O_ domain by generating torque and allowing conformational changes in F_1_ which leads to ADP phosphorylation to yield ATP. In mitochondria, the bi-functional enzyme can also work in reverse when the Δ*p* drops. In this case, ATP hydrolysis by the F_1_ domain fuels H^+^ pumping by F_O_ which re-energizes the mtIM [133]. This energy-dissipating mechanism has also been associated with a variety of pathological conditions [134]. Both in the forward and in the reverse reaction, H^+^ translocation across the mtIM is due to the reversible protonation/deprotonation of carboxylic sites, which also converts H^+^ flux into F_O_ rotation. The disconnection between these two matched activities, namely H^+^-translocation and catalysis, “uncouples” the F-ATP synthase and often leads to mitochondrial dysfunctions. The enzyme sensitivity to the antibiotic oligomycin, which by binding to the enzyme covers the H^+^ binding sites and blocks the *c*-ring rotation and enzyme catalysis in either directions [135], is taken as a parameter of the efficient coupling between the two domains.

This amazing rotary enzyme complex offers a good example of how molecular structure and function are tightly linked features [130], which cannot work without each other.

### 4.2. Structural Basis of the Bioenergetic Mechanism

Among all F-ATPases, the mitochondrial enzyme complex has the most complex subunit composition [136]. F_1_ protrudes in the mitochondrial matrix as an asymmetric hexagonal globular assembly of α and β subunits, arranged as (αβ)_3_ around the γ subunit of central stalk which connects F_O_ to F_1_. The hexamer, in which α and β subunits alternate, hosts three catalytic and three non-catalytic sites. The three catalytic β-sites can adopt three conformations, namely “open”, “closed”, and “semi-closed”, defined as β_E_, which is empty, β_TP_, which contains Mg-ATP or Mg-ADP and β_DP_, which hosts Mg-ADP. Each of the three non-catalytic α subunits binds Mg-ATP [137] and also Mg-ADP [128]. According to the binding change mechanism, the β conformations interconvert each other as the *c*-ring rotates and transmits the rotation to central stalk [138], which when it meets the catalytic sites, changes their conformation and affinity for nucleotides. So, during a complete turn (360°) each catalytic site undergoes (stepwise) all the three different conformations leading to synthesize/hydrolyze three ATP molecules. Moreover, Mg-ATP bound to the non-catalytic sites allows ADP release from the βDP site and removes the Mg-ADP-driven enzyme inhibition during multiple ATP hydrolysis turnover [139]. Interestingly, the mammalian γ subunit shows two isozymes, a “heart isoform” and a “liver isoform”, which differ by a single amino acid, an additional aspartate at the C-terminus of the liver enzyme. The former is expressed in tissues with a high and variable energy demand such as heart, muscle, diaphragm, while the latter is expressed in brain, thyroid, spleen, pancreas, kidney, testis and liver, which, apart from brain, show a low ATP consumption and a steady energy demand. Even if the C-terminus Asp in the liver γ-subunit may form salt linkages and decrease the enzyme efficiency, no differences in enzyme kinetics were found, so the two γ isoforms probably only have a regulatory function [140].

Other than by the γ subunit, the functional coupling of the two main sectors F_O_ and F_1_ in mammalian mitochondria is ensured by the peripheral stalk and especially by OSCP [134].

The membrane-embedded F_O_ domain, which allows H^+^ flow across the mtIM, hosts the *a* subunit, which channels H^+^, and the *c*_n_-ring, namely the rotor, which consists of a cylindrical palisade of n subunits. The number n varies in the range 8–15 among the species, is constant in the same species and determines the ATP bioenergetic cost, which is obtained from the ratio between the number of translocated H^+^ and the three ATP molecules constantly produced in a complete (360°) rotation of the rotor. Accordingly, the number of transported H^+^ per complete rotation of the *c*-ring is related to the number of H^+^ binding site(s) on each *c* subunit. In general, an increase in the *c*-ring size and consequently in the number of H^+^-binding *c*-subunits, implies an increase in the bioenergetic cost of ATP, since more H^+^ should flow downhill across the mtIM to produce one ATP molecule. Mammals which show a small *c*-ring (n = 8) are highly efficient by dissipating less H^+^ to yield one ATP, namely they obtain ATP at a low bioenergetic cost [141]. So, the number of *c* subunits, and of H^+^ binding sites, can be taken as a rotor efficiency parameter. In the OXPHOS system the transmembrane driving force Δ*p* which links substrate oxidation to ADP phosphorylation to yield ATP, consists of an electrical ΔΨ (membrane potential difference) and of a chemical component, ΔpH (pH gradient) between the positive and negative side of the mtIM. During evolution, species have adapted the *c*-ring size to the prevailing electrochemical parameter, ΔΨ or ΔpH. Accordingly, when ΔΨ prevails and the pH gradient is low, the *c*-ring is small [142]. This is the case of mammals, generally vertebrates and some invertebrates [143]. Conversely, a prevailing ΔpH is associated with the large *c*-rings (n = 11–15), typical of chloroplasts and bacteria [144,145].

The H^+^ pathway within F_O_ only recently has been elucidated. It is built step by step within the *a* subunit by selected amino acid side chains, which form a quite unexpected route along the mtIM allowing membrane crossing in a perpendicular way to the two aqueous half-channels. The two half-channels are discontinuous and open one in the matrix and the other in the lumen and both contain two horizontal highly conserved α-helices, named H5 and H6, which lie along the mtIM and are juxtaposed to the *c* subunits, thus providing access to the *c*-ring H^+^ binding sites [146]. This α-helix arrangement in the mtIM features all rotary A-, F- and V-type ATPases [147]. Each *c* subunit is hairpin-shaped and spans the mtIM. These gathered hairpins form a sort of hourglass, seen laterally from the membrane side, whose concavity hosts the H^+^-binding sites on the outer C-terminal α-helices of *c* subunits. Therefore, the horizontal α-helix arrangement perfectly fits the rotor concavity [148].

The H^+^ transfer across the mtIM in either direction requires subsequent protonation/deprotonation steps of H^+^ binding sites that consist of amino acid side chains that interconvert between the protonated and deprotonated states. Interestingly, in mitochondria at the edge of the *cristae*, the pH value is higher than that in the IMS opposing the inner boundary membrane (IBM), as well as the *cristae* ΔΨ generated by the respiratory chain [149] is higher than in the IBM [150]. The H^+^ flow through F_O_ for ATP synthesis occurs from the lumen or *intracrista* space of the half-channel formed by the hydrophilic cavity created by the end of H5 and H6 helices of *a* subunit, and H3 helixes of *b* and *f* subunits. The H^+^ pathway starts from *a*His-168 and *a*His-172 on H5 helix and ends at *a*Glu-203 on H6 helix of *a* subunit, which acts as intermediate H^+^-donor to *c*Glu-58 of the *c*-ring [128,151]. The Glu-His interactions establish multiple H-bridges in the membrane half-channel that, by changing the carboxylic group p*K*_a_, allow its protonation. On the opposite half-channel on the luminal side viewed from the *intracrista* space, the *a*Glu-145 on H5 (matrix) helix acts as “de-protonator” which restores the original p*K*_a_ of *c*Glu-58 allowing the H^+^ detachment. The half-channel arises where a conserved *a*Pro-153 bends the *a*H5 helix [151]. Then, the folding is enforced by DAPIT subunit and H4 helix of *a* subunit in order to make the half-channel hydrophilic core conformationally adapt to the *c*-ring shape (Figure 4).

During the H^+^ translocation through the membrane-embedded F_O_ domain, a putative participation of hydronium ion (H_3_O^+^) was also suggested [152]. Accordingly, in prokaryotes and eukaryotes the bell-shaped pH profile of the F_1_F_O_-ATPase inhibition by dicyclohexyl-carbodiimide (DCCD), which covalently binds to *c*-subunit carboxylates, is not compatible with carboxylate protonation of H^+^ binding sites, but it may be explained by H_3_O^+^ coordination [143,153]. Indeed, on considering the position of molecules able to establish hydrogen bonds in the half-channels, whose core is hydrophilic, H^+^ translocation may be coordinated by water molecules by the Grotthuss mechanism, namely by proton jumping from covalent to H-bonds and vice versa [128,154]. During ion translocation, the *c*Glu-58 adopts different conformations, namely the H^+^-unlocked anionic form (carboxylate) to face the aqueous environment of the half channels, and the H^+^-locked form of the protonated carboxylic group to face the hydrophobic environment of the mtIM. In the locally hydrated luminal half-channel, the *c*Glu-58 carboxylate is oriented in an outward-facing (H^+^-unlocked) conformation before protonation and turns to an inward-facing closed (H^+^-locked) conformation when protonated. On the opposite half-channel, which faces the matrix, the *c*Glu-58 in the H^+^-locked conformation turns to the H^+^-unlocked conformation after deprotonation. The *c*-ring key carboxylates embedded in the mtIM are always in the H^+^-locked conformation to enter the lipophilic mtIM [155] and make the *c*-ring rotate due to Brownian motion [131]. The rotation direction when viewed from F_O_ toward F_1_ is clockwise during ATP synthesis and counterclockwise during ATP hydrolysis [133,156]. The low pH in the luminal half-channel favors *c*Glu-58 protonation and the consequent locked conformation, pushed by Δ*p* that favors the entry to the mtIM. When an almost entire rotor rotation is completed, the *c*Glu-58 reaches the half-channel on the matrix side where the low H^+^ concentration as well as the negative charge of the “H^+^-releasing site” on H5 helix of *a* subunit, favors its deprotonation associated with the H^+^-unlocked carboxylate form. The positive charge of a crucial arginine, *a*Arg-159, acts as electrostatic barrier between the two half-channels, prevents H^+^ short circuiting and attracts the negatively charged *c*Glu-58 carboxylates [157]. Moreover, *a*Arg-159 helps the *c*Glu-58 de-protonation during H^+^ translocation and prevents salt bridges between *a* and *c* subunits which would block the rotation. Interestingly, *a*Arg-159 and the H^+^-unlocked *c*Glu-58 are too far (about 4.5 Å) and cannot interact [144], so the two half-channels for H^+^ entry and exit at the *a*/*c*-ring interface are spatially offset to allow the rotation direction adjust to Δ*p* [135].

During ATP hydrolysis the transition between the three β conformations in F_1_ drives the rotations of the γ subunits and of the *c*-ring and increases the Δ*p* by pushing H^+^ uphill [134].

The coupling of energy transduction and H^+^ translocation is allowed by conformational adaptations of the F_1_ and F_O_ domains [158] that adapt to each other, helped by flexible regions in the peripheral stalk. The latter undergoes torsion during catalysis, but also the *c*-ring shows conformational fluctuations when it contacts the *a* subunit. Therefore, coordinated conformational changes within the enzyme proteins produce and synchronize the torsion.

### 4.3. The ATP Synthase Supramolecular Arrangement and the Mitochondrial Shape

The mitochondrial F_1_F_O_-ATPase functions are ensured by the enzyme supermolecular arrangement. By self-assembling in dimers and tetramers, the F_1_F_O_-ATPase rules most bioenergetic and structural functions in eukaryotic mitochondria [159]. Accordingly, the occurrence of *sms*, which lack in bacteria and chloroplasts, in F_O_ is linked to the *sms* involvement in supercomplex arrangement and in the mtIM ultrastructure [130]. Consistently, no F_1_F_O_-ATPase dimers were found in prokaryotes and chloroplasts, while some differences between yeast and mammalian F_1_F_O_-ATPases occur in the building of contact sites. It seems to have been ascertained that, while the basic subunits allow the overall functionality of the enzyme complex, these *sms* are required to join the monomers and to bend the membrane. The *sms* were differently named according to the taxa: A6L, *e*, *f*, *g*, and DAPIT, which corresponds to the *k* subunit in yeasts, and mammalian 6.8PL, which corresponds to the orthologue *i/j* in yeasts [128]. The membrane subunits are mainly encoded by nuclear genes, apart from *a*, A6L and *c* subunit (identified also as *6*, *8* and *9* subunit) for yeast F_1_F_O_-ATPase and only *a* and A6L in mammals which are encoded by the mitochondrial DNA. Notably, different *sms* structurally contribute to the F_1_F_O_-ATPase dimerization in yeasts and in mammals [127,160]. Three different isoforms occur in the mammalian *c* subunit.

F_1_F_O_-ATPase monomers form dimers arranged in long rows on the mtIM and MICOS (mitochondrial contact site and *cristae* organizing system) complex at *cristae* junctions cooperate to yield the *cristae* morphology in agreement with their opposing effects on membrane curvature [161,162,163]. Moreover, recent work shows that the F_1_F_O_-ATPase dimerization factor subunit *e* interacts with the MICOS component Mic10, which stabilizes the enzyme oligomers and possibly temporal orchestration of cristae biogenesis. Moreover, the F_1_F_O_-ATPase dimerization and the MICOS component Mic60 induce opposite membrane curvature, namely the former promotes a negative curvature and the latter a positive curvature (concave and convex as viewed from the matrix, respectively). Most likely, the interaction between MICOS and dimeric F_1_F_O_-ATPase, which implies a high mobility of the enzyme monomers in the *crista* membrane, enables spatial and temporal coordination during *crista* biogenesis and dynamics [164].

Interestingly, the dimerization interface in different species shows different subunit composition with no apparent homology [136].

Four different types of F_1_F_O_-ATPase dimers were described in mitochondria. In mammals (e.g., *Bos taurus*, *Sus scrofa domesticus,* and *Ovis aries* heart) and yeasts (*Saccharomyces cerevisiae*) V-shaped dimers, named type I dimers, are localized on the rim of the *cristae* with an mtIM convexity of about 90° [137,149]. Type II, III and IV dimers, which differ in the angle between monomers, were only found in algae, protozoa and small invertebrates [148,159].

Mammalian F_1_F_O_-ATPases can also form a H-shaped tetramer, which in turn consists of two assembled type I dimers that lie antiparallel to each other. The dimers are joined by two IF1 subunits, an endogenous protein inhibitor [127], evolutionary conserved throughout eukaryotes, which only blocks ATP hydrolysis but not ATP synthesis. Most likely, IF1 normally is not essential for cell survival, but can be crucial under pathological or stress conditions [165], as suggested by its overexpression associated to the decrease in OXPHOS in some cancer types [166]. However, IF1 not only prevents ATP dissipation by ATP hydrolysis, but also contributes to mitochondrial morphology [127,167,168]. The IF1 dimer, which becomes active at acidic pHs in the matrix and stems from the dissociation of the tetramer, inhibits ATP hydrolysis when Δ*p* collapses. In the tetramer, IF1 links two F_1_ domains of two opposite dimers; it binds to the catalytic interface between the α_DP_ and β_DP_ subunits in loose binding conformation. Therefore, the two monomers of the laterally opposite dimer are connected by the same IF1; these two IF1 proteins make the tetramer steady and block ATP hydrolysis by a ratchet-like action on the rotor. The F_1_F_O_-ATPase dimers are also maintained in rows by a long-range attractive force that stems from the relief of the overall elastic strain of the mtIM [169]. Other than by interactions with IF1 dimers, the mammalian tetramer is maintained by *e*-*e* and *g*-*g* interactions between two opposed and diagonally arranged monomers [170]. Moreover, the monomers are joined side-by-side in a row through DAPIT interaction with *a* subunit at the matrix space and with *g* subunit both at the matrix side and the intra*crista* space. The dimers are linked by interactions of one monomer 6.8PL with *f* and *e* subunits of the other monomer, and vice versa, at the matrix and intra*crista* space, respectively. In addition, the subunits *a*-*a* in the middle of mtIM, the *f*-*f* at the matrix and *e*-*e* at the intra*crista* space establish multiple contacts between each monomer pair [128].

Moreover, the association of two F_O_ domains depends on *e* and *g* subunits and on a putative conserved dimerization GxxxG motif in the transmembrane α-helices of both subunits [171,172]. The substitution of a glycine residue by leucine into the *e* subunit motif led to the loss of *g* subunit, destabilized the dimer and resulted in an onion-like structure in the *cristae* [171]. The individual α-helix of *e* subunit and the H3 of *g* subunit interact with their respective GxxxG motif and joined to H2 of *b* subunit form the triple transmembrane helix bundle (TTMHB). The TTMHB tilt is driven by a U-turn structure formed by H2 and H3 helices of *b* subunit. This subunit assembly creates a reminiscent BAR-like domain that bends the mtIM [128,173] to form the apex of *cristae*, but *e* and *g* subunits also keep the monomers together. Accordingly, disulfide cross-link experiments showed the interactions between *e*-*g* or *e*-*e*/*g*-*g* increase the stability of F_1_F_O_-ATPase dimers and oligomers, respectively, in mitochondrial digitonin extracts [172].

In mammals, subunit–subunit interactions produce the F_1_F_O_-ATPase supermolecular arrangement, in which monomer pairs form dimers and dimer pairs form tetramers, which in turn form long rows of oligomers. This structural arrangement produced by subunit–subunit interactions, also forces the mtIM to be convex at the apex of the *cristae*. The enzyme subunits involved in oligomerization, though not directly involved in catalysis, contribute to the maintenance of *cristae* architecture, which is essential for an efficient respiration by acting as proton sink. A mutation in subunit *k*/DAPIT results in aberrant *cristae* formation, defective F_1_F_O_-ATPase dimerization and a mild Leigh syndrome phenotype [164]. *Cristae* remodeling and F_1_F_O_-ATPase dimerization may be also involved in cell differentiation [174].

Interestingly, in some forms of cancer, the overexpression of IF1, which leads to the F_1_F_O_-ATPase dimerization and *cristae* formation, could contribute to neoplastic degeneration and evasion of apoptosis. The absence of IF1 in Luft’s disease, a rare mitochondrial disease, is associated with densely packed mitochondrial *cristae* [175]. Moreover, the acidic phospholipid CL is critical for oligomerization of F_1_F_O_-ATPase multimeric complexes either by direct interaction with the enzyme or by inducing the proper membrane curvature. CL, acting like a non-bilayer forming phospholipids, interacts with the F_1_F_O_-ATP synthase and may reduce the free energy of the extreme mtIM curvature to stabilize high-curvature folds [164,176]. From these few examples it seems clear that any change in at least one among the multiple factors which rule the mitochondrial morphology can easily lead to pathology. The F_1_F_O_-ATPase can work in monomeric and dimeric form; however, the loss of the supramolecular organization of the F_1_F_O_-ATPase causes aberrant membrane morphology results in respiratory defects.

On these bases CL defects, as in Barth Syndrome (see Section 3.1.3) can affect the supramolecular arrangement of the F_1_F_O_-ATPase, the mitochondrial shape and, as depicted in the following section, the mitochondrial permeability transition.

During aging, the typical rim shape of the *cristae* disappears and the F_1_F_O_-ATPase dimers dissociate into monomers. Consistently, the mitochondrial morphology changes, showing a vesicular mtIM, leading to mitochondrial dysfunction and cell death [177]. Even if most results come from in vitro studies on experimental models and should be deepened and confirmed in vivo, recent advances suggest that mitochondrial changes are a driving force, rather than a consequence, of the aging process and neurodegeneration [176]. The connection among the F_1_F_O_-ATPase supramolecular arrangement, the mitochondrial shape and some human diseases, as pointed out by these few examples, is also made clearer by the enzyme involvement in the mitochondrial permeability transition pore (mPTP), the topic of the following section.

### 4.4. How the ATP Synthase/Hydrolase Is Involved in the Mitochondrial Permeability Transition Pore

In recent years, the F_1_F_O_-ATPase has been implicated in the mPTP, a large pore in the mtIM, which permeabilizes the mtIM to ions and other solutes [178,179,180]. As the membrane becomes permeable, the massive water influx in the matrix results in mitochondrial swelling. For short full openings, the mPTP is reversible, but prolonged openings are irreversible and trigger the release of cyt. *c* and other pre-apoptotic factors which drive the cell to death [181]. The mPTP formation is stimulated by ROS and Ca^2+^ increase, by the binding of the mitochondrial protein cyclophilin D (CyPD) in mitochondria and inhibited by H^+^, Mg^2+^, adenine nucleotides, and cyclosporin A (CsA), which displaces CyPD and desensitizes the mPTP to Ca^2+^ [182]. The mPTP activity mainly appeared as a property of mammalian and yeast mitochondria, where it was intensively investigated. However recently the mPTP was also reported in mussel [183], in sea urchin oocytes [184], in the nematode *Caenorhabditis*
*elegans* [185], in plathelminths [186] and insects [187,188], while it is still controversial in crustaceans [189,190]. Indeed, it is not a “vertebrate invention” to rule cell fate, as initially thought [191].

At present, the problem of the involvement of F_1_F_O_-ATPase in the mPTP and especially its structural participation in the mechanism of mPTP formation and opening is near to being solved [128]. This quite surprising task of the F_1_F_O_-ATPase, which has been supposed by different research groups, points out the versatility of this intriguing enzyme complex, which emerges as the playmaker of cell life and cell death. Two main F_1_F_O_-ATPase sites have been involved in the mPTP formation: the *c*-ring [179,180] or the monomer–monomer interface of the dimer [178].

The hydrolytic function of the F_1_F_O_-ATPase, namely ATP hydrolysis, can be sustained by metal divalent cations different from the natural cofactor Mg^2+^, such as Ca^2+^, which is unable to activate the enzyme complex in the opposite function of ATP synthesis [192,193,194,195]. Interestingly, Ca^2+^ rise in the mitochondrial matrix initiates a cascade of events which lead to cell death. So, in recent times Ca^2+^ binding to the F_1_F_O_-ATPase in replacement of Mg^2+^, an event which can easily occur in the presence of relatively high Ca^2+^ concentrations in mitochondria, since the enzyme affinity for Ca^2+^ is lower than that for Mg^2+^ [193,196], has been associated with the mPTP opening [192,195,197,198,199].

Accordingly, the structure of mammalian F_1_F_O_-ATPase exposed to Ca^2+^ and revealed by cryo-electron microscopy, shows unusual states, not identified when Mg^2+^ acts as cofactor, which can be ascribed to mPTP opening. Indeed, in the new “bent-pull” model, the Sazanov lab highlights the role of *c*-ring and of some *sms* in mPTP formation. Accordingly, when Ca^2+^ replaces Mg^2+^ in the catalytic binding site, the cation insertion in the Ca^2+^-activated F_1_F_O_-ATPase, due to the higher steric hindrance, would promote conformational changes of F_1_ domain, in turn transmitted by the peripheral stalk [198] to membrane subunits that form the mPTP [198]. Notably, when the Ca^2+^-activated F_1_F_O_-ATPase activity is decreased by various inhibitors, the mPTP formation is delayed or even prevented [196,197,200]. The *c*-ring contains two different phospholipids at the two opposite sides of its cavity. At the matrix side, phosphatidyl serine is anchored by ionic interactions to the positive charge of Arg-38 of *c* subunits, while at the intra*crista* side, Lys-71 of *e* subunit coordinates a lyso-phosphatidylserine. Since the phosphatidylserine double acyl-chain does not have enough space around the *c*-ring plug and is linked to the *c*-ring, it rotates together with the rotor. Conversely, the monoacyl chain of lyso-phosphatidylserine acts as “lubricated” lipid plug. The two lipids inside the *c*-ring are separated by the conserved *c*Val-16 [128]. The F_1_F_O_-ATPase distortion and tilt induced by Ca^2+^ trigger changes in the conformational states of the Ca^2+^-activated F_1_F_O_-ATPase that open the mPTP. Most likely, the signal propagation from F_1_ to F_O_ through the long helix of *b* subunit modifies the TTMHB assembly and changes the position of *e* subunits, which expel the lyso-phosphatidylserine from the central hole inside the *c*-ring and opens the channel at the positive mtIM side [181]. Consequently, the curved *crista* ridges and the F_1_F_O_-ATPase dimerization are lost [177,199,201]. According to Sazanov’s hypothesis, the water molecules inside the *c*-ring destabilize the phosphatidylserine which pushes out the lipid plug and creates a pore through the *c*-ring, while the consequent conformational change detaches F_1_ from F_O_. Thus, structural/conformational F_1_F_O_-ATPase changes are involved in the (ir)reversible mechanism of the mPTP, which opens and closes and rules cell fate. The most recent data strongly sustain the hypothesis that these changes mainly involve the *c*-ring, which emerges as the main character in mPTP formation (Figure 5).

However, this matter is still hot and some debate on this topic remains. The possibility that differently sized pores can be formed and coexist in the mtIM satisfactorily combines the conclusions drawn from different experimental approaches. Accordingly, the mtIM could be depolarized by any increase in conductance upon mitochondrial Ca^2^^+^ overload, due to transmembrane channels and/or transporters [202]. In this scenario, the Ca^2+^-activated F_1_F_O_ATPase would contribute to the membrane depolarization by inducing the largest mPTP pore. It seems reasonable to think that smaller sub-conductance activities, which contribute to the mPTP can be ascribed to many other mitochondrial channels/transporters [203]. However, the mitochondrial F_1_F_O_-ATPase remains the most likely candidate as main high-conductance major channel or mPTP by definition, proven to be inhibited by CsA, but not by bongkrekate (BKA), which is known as inhibitor of the adenine nucleotide translocase [202,204].

Moreover, the adenine nucleotide translocase isoforms could form a secondary low-conductance mPTP inhibited by both CsA and BKA [205]. Notably, the mPTP activity can be enhanced by Ca^2+^ and pharmacologically modulated by selective inhibitors of F_1_ [196] and F_O_ domain [206,207,208]. Newly, purified and functionally active F_1_F_O_-ATPase monomers [209] and dimers [210] act as a voltage-gated ion channel endowed with mPTP-like properties.

Dysregulated mPTP opening is involved in mitochondrial dysfunctions which feature a variety of diseases such as neurological and cardiovascular disorders, type 1-diabetes [211], cancer [212], inflammatory bone diseases [213] and diseases due to the exposure to contaminants [214]. In general, pathological conditions associated with oxidative stress also involve mPTP dysregulation [182]. Notably, the mPTP has been also involved in lifespan [215] and bone repair [216]. So, the discovery and design of mPTP rulers is at present a great challenge in pharmacology [217,218,219]. Recently, the use of natural products such as mPTP modulators raised a great interest in ethnomedicine [220].

## 5. SCs Association Protects from ROS Damage

The supramolecular organization of the respiratory chain has revealed a hitherto unknown link between mitochondrial structure and generation of the so-called ROS. In this section we will provide evidence that there is a biunivocal relation between ROS and SC association, i.e., SC association protects mitochondrial structure from ROS damage, but at the same time the SC association limits ROS generation from the respiratory chain.

ROS, in the past considered exclusively as major factors for cellular toxicity, are recognized today to be important physiological factors involved in cell signaling, by affecting the redox state of signaling proteins [221].

ROS is a collective term including oxygen derivatives, either radical or non-radical, that are oxidizing agents and/or are easily converted into radicals. Transition metals such as iron and copper, when in a free state, have a strong capacity to reduce O_2_. Stepwise addition of an electron generates in sequence the superoxide radical anion O_2_^−^, the peroxide ion (which is a weaker acid and is protonated to hydrogen peroxide H_2_O_2_), and the hydroxyl radical OH. The hydroxyl radical is extremely reactive with a half-life of less than 1 ns; thus, it reacts close to its site of formation.

### 5.1. Mitochondrial Sources of ROS

In the cell environment, ROS can stem from exogenous and endogenous sources [222]. Exogenous sources of ROS include UV and visible light, ionizing radiation, drugs and environmental toxins. On the other hand, endogenous sources of ROS embrace enzyme activities such as: xanthine oxidase, cytochrome P-450 enzymes in the endoplasmic reticulum, peroxisomal flavin oxidases and plasma membrane NADPH oxidases. However, the membrane-ETS in the mtIM is probably the major source of ROS. Other mitochondrial enzyme systems can significantly contribute to ROS generation [223], as dihydrolipoamide dehydrogenase (a subunit of the α-ketoglutarate and pyruvate dehydrogenase complexes), monoamine oxidase, mitochondrial nitric oxide synthase and the adaptor protein p66^Shc^. These systems are not relevant to this review and will not be considered here.

#### ROS from the Respiratory Chain

In mammalian mitochondria, and particularly in the respiratory chain and related enzymes, at least ten different sites of superoxide/H_2_O_2_ production have been identified [224]. The FMN and CoQ-binding sites of CI and the Q_O_ site (at the outer or positive side) of CIII are considered as the most important sites for superoxide production. However, other sites have also been defined (Figure 6).

Most of superoxide is generated at the matrix side of the mtIM, as superoxide is detected in SMP, which are inside-out vesicles, namely have an opposite orientation with respect to mitochondria. However, a significant amount of superoxide is formed at the outer face of the mtIM [225]. It is likely that CI releases ROS in the matrix, while CIII releases ROS mostly in the IMS. The superoxide released at the IMS may be directly exported to the cytoplasm through an anion channel related to the Voltage-Dependent Anion Channel, VDAC.

CI is a major source of superoxide production in different mitochondrial types. Several prosthetic groups in CI were proposed to directly reduce oxygen, including FMN [226,227,228], ubisemiquinone [229,230], and iron sulfur (FeS) cluster N2 [231,232,233].

Grivennikova and Vinogradov [234] observed both superoxide and hydrogen peroxide generation by CI and attributed superoxide formation to FeS cluster N2, an hydrogen peroxide formation from 2-electron oxidation of fully reduced FMN. Rotenone, a specific inhibitor of CI, enhances ROS formation during forward electron transfer in the respiratory chain [231].

The highest rate of ROS generation in isolated mitochondria occurs during oxidation of succinate [235]. Since this ROS production is inhibited by rotenone [229,236,237], it is commonly attributed to CI during the energy-dependent reverse electron transfer from succinate to NAD^+^ through CoQ [238,239,240]. See also Section 3 for the role of the reduced CoQ pool in reverse electron transfer and ROS-dependent CI destruction (cf. Guáras et al. [120]).

The formation of superoxide in CIII depends on its peculiar mechanism of electron transfer, the so-called Q-cycle [241]. If the lifetime of the semiquinone (Q) at the outer Q_O_ site, is prolonged, as in the controlled state when the electric potential is high, it reacts with oxygen, thus forming superoxide [242,243]. This conclusion is also reached by the superoxide increase induced by the specific CIII inhibitor, antimycin [244].

More recent studies, however, suggest that the oxygen reductant is the semiquinone formed in the so-called semi-reverse reaction in which cytochrome *b*_L_ reduces the fully oxidized quinone [245,246]. Osyczka [247,248] proposes that a metastable radical state, nonreactive with oxygen, safely holds electrons at a local energetic minimum during the oxidation of ubiquinol. This intermediate state is formed by interaction of a radical with a metal cofactor of a catalytic site, presumably the FeS cluster, under physiological conditions.

The other sites in the respiratory chain which may be a source of ROS (e.g., CII, glycerol phosphate dehydrogenase, dihydroorotate dehydrogenase, ETF and ETF dehydrogenase) are reviewed in [249].

### 5.2. Control of ROS Production

The steady-state concentration of ROS in a cell depends on many factors, including the rate and site of production, their lifetimes and diffusion constants, the interconversions between different ROS, and the removal rate by different antioxidant systems.

The ROS generation by isolated mitochondria accounts for 0.1–0.2% of oxygen consumed, but the rate and extent vary among different tissues and substrates [250,251].

The mitochondrial ROS level is physiologically regulated, as expected for their role as cellular signaling molecules (cf. Section 5.3).

Mitochondrial ROS production increases in State 4 (controlled non-phosphorylating state) when the membrane potential is high and the electron transfer rate decreases [252]. In fact, under this condition the respiratory carriers capable of donating electrons to oxygen are more reduced. For this reason, uncoupling may limit the production of ROS by releasing excessive membrane proton potential. In rat hepatocytes, a futile cycle of H^+^ pumping and proton leak may account for 20–25% of respiration [253] and even more in perfused rat muscle. Uncoupling may be achieved by activating proton leak through uncoupling proteins (UCP) [244] (for a different role postulated for uncoupling proteins, cf. Wojtczak et al. [254]). In this way, although a tissue may dissipate a conspicuous part of the energy conserved by its mitochondria, the mitochondrial respiratory chain is maintained under more oxidized conditions, thus preventing the overproduction of damaging species.

UCP2 structure supports a fatty acid cycling mechanism for uncoupling [255]: UCP2 expels fatty acids in their anionic form from the negative matrix, while the protonated acids flow back from the positive exterior by passive diffusion through the bilayer. The resulting effect is H^+^ back diffusion to the matrix, so that Δ*p* dissipation decreases superoxide formation. UCP-mediated antioxidant protection and its impairment are expected to play a major role in cell physiology and pathology [256]. Mitochondrial uncoupling could be exploited to treat human diseases, such as obesity, cardiovascular diseases, or neurological disorders [257] (cf. Section 6).

Besides high membrane potential, other reasons may induce a decrease in the rate of electron transfer in CI leading to overproduction of ROS: for example, subunit phosphorylation that inhibits CI activity, enhances its ROS generating capacity [258,259,260]. Hence, endocrine alterations may affect ROS formation by changing the phosphorylation state of the complex.

Kadenbach et al. [261] proposed a ROS-generating mechanism that depends on regulation of cytochrome *c* oxidase. The cAMP-dependent phosphorylation of subunit I of the oxidase, by increasing its sensitivity to allosteric ATP inhibition, decreases H^+^ translocation by the enzyme, with resulting decrease in mitochondrial membrane potential and, consequently, in ROS generation by the respiratory chain. On the contrary, dephosphorylation of CIV under stress conditions would induce increase in membrane potential and a burst of ROS generation by mitochondria. The discovery of a mitochondrial cAMP signalosome [262], modulating the allosteric inhibition of CIV by ATP, underlies a universal mechanism for metabolic regulation in eukaryotes.

In addition, cyt. *c* is regulated by different mechanisms, including post-translational modifications [263,264]. Specific cyt *c* residues may be phosphorylated in a tissue-specific mode. These modifications adjust electron flux and membrane potential to minimize ROS production under normal conditions. Conversely, under pathologic and acute stress conditions, such as ischemia-reperfusion, the dephosphorylation leads to membrane hyperpolarization, excessive ROS generation, and cyt. *c* release.

Other protein modifications in the respiratory complexes can modulate ROS production [265].

### 5.3. ROS as Signals

It is now established that ROS can act as second messengers [265,266,267,268,269,270,271,272,273,274] by modulating the expression of several genes involved in signal transduction. The effects related to ROS signals are different and even opposite, depending on the concentration. ROS may be oncogenic and promote proliferation, invasiveness, angiogenesis and metastasis [275,276], and accordingly ROS generation increases in various cancers. Nevertheless, ROS can also promote anti-tumorigenic signaling and trigger oxidative stress-induced tumor cell death.

Many of the ROS-mediated cellular signals paradoxically protect the cell against oxidative stress, but above a given threshold ROS become harmful and induce oxidative stress [268]. Not all ROS are equally suitable for signal transduction: the OH· radical is too aggressive to participate in catalyzed reactions, whereas O_2_^−^ and H_2_O_2_ are known as important signaling molecules. Moreover, other reactive species are involved in redox signaling, such as nitric oxide, hydrogen sulfide and oxidized lipids [274].

Because some ROS such as H_2_O_2_ are oxidants, they modify the redox state of signaling proteins by oxidizing specific sulfhydryl groups [274,277,278] in enzyme-catalyzed reactions.

Several transcription factors contain redox-sensitive cysteine residues at their DNA-binding sites [279,280]. The ensuing formation of disulfide bonds acts as a redox switch to transduce the response of the protein to redox signaling [271,281].

Following the oxidation of signaling proteins by ROS, other post-translational changes (e.g., phosphorylation, acetylation, ubiquitination, and SUMOylation [270]) may occur in the same protein and in other proteins of the signaling cascade.

Nrf2 is an important example of mitochondrial ROS signaling which leads to nuclear gene expression changes [282]; in the presence of ROS, NRf2 is transferred from the cytoplasm to the nucleus, where it binds the DNA antioxidant response element of genes involved in the antioxidant response (e.g., heme oxygenase, NRF-1 and other inducers of mitochondrial biogenesis) [283].

Signaling proteins modified by ROS include phosphoprotein phosphatases (PTPs), Ras, large G-proteins, serine/threonine kinases of the MAPK families, transcription factors as AP-1, NFκB, p53 and others. A different effect is exerted on these proteins: PTPs are inhibited, while nuclear transcription factors are activated [284]. Different combinations of activation or deactivation of these proteins may drive the cell to either survival or death.

Mitochondrial ROS production is involved in preconditioning, by which low doses of a noxious stimulus promote a protective response against future injury [285,286,287]. Some protective pathways converge on the inhibition of mPTP opening during reperfusion [287].

Mitochondrial ROS are also important signals which rule the inflammation by activating the inflammasome [288,289]; treatment with antioxidants could scavenge excessive ROS and attenuate inflammatory responses by suppressing inflammasome activation [290]. ROS also regulate autophagic processes including mitophagy [291].

#### 5.3.1. Regulation of Mitochondrial ROS in Cell Signaling

Mitochondrial ROS levels depend on the dynamic equilibrium between their rates of production and removal.

Studies on ROS generation under different substrate, inhibitor conditions and different oxygen tensions in rat liver mitochondria lead to the conclusion that only CI may be a significant source of ROS at physiological O_2_ concentration [292]. The factors directly associated with respiratory activity (e.g., the redox potential of the NAD^+^/NADH couple and the proton-motive force) regulate the reduction in oxygen [268]) but these factors are in turn dependent on other parameters (cf. also Section 5.2).

Moreover, mitochondria enhance ROS generation in response to external stimuli, such as TNFα [293], hypoxia ([294], cf. Section 5.3.2), serum deprivation [295] and oxidative stress itself (ROS-induced ROS release, Zorov et al. [296]). Several proteins, such as p53, p66^Shc^, the Bcl-2 family and Romo-1 [267] control mitochondrial ROS generation.

An interplay of mitochondria with non-mitochondrial ROS generating systems also occurs, as with the NADPH oxidases (NOX) family [297,298,299].

On the other hand, control of ROS signaling may be operated by their removal by a series of enzymatic systems [300], including various isoforms of superoxide dismutase, catalyzing the dismutation of superoxide, whereas catalase, glutathione peroxidase and peroxiredoxins remove H_2_O_2_. In addition, an integrated set of thiol systems in mitochondria prevents oxidative damage [271] and may transduce redox signals.

Peroxiredoxins (Prxs) are small dithiol proteins constituting a major family of peroxidases (six isoforms in mammals). Due to their high sensitivity to oxidation by H_2_O_2,_ Prxs are sensors and transducers of H_2_O_2_ signaling [301] by transferring their oxidation state to effector proteins. The mitochondrial matrix contains large amounts of Thioredoxin-2 [302,303] and Prx3 [304].

The appropriate mitochondrial redox environment is greatly ascribed to glutathione [305,306,307] which is abundant and versatile to counteract hydrogen peroxide and lipid hydroperoxides, being a cofactor of glutathione peroxidase or glutathione-*S*-transferase. The mitochondrial GSH:GSSG ratio generally exceeds 100 and is widely used as an indicator of the redox status [308].

Protein glutathionylation is a regulatory mechanism that occurs through thiol disulfide exchange by protein disulfide isomerase [309,310]; the primary sistem catalyzing de-glutathionylation of protein-mixed disulfides with glutathione is glutaredoxin [311]. Mitochondrial proteins are highly susceptible to reversible *S*-glutathionylation, a post-translational modification which is favored by the unique physico-chemical properties of the mitochondrion [312], which contains a variety of *S*-glutathionylation targets playing important physiological roles: if these reactions are defective, pathological consequences are expected to occur.

#### 5.3.2. Hypoxia and ROS Production

Among the many signaling functions, ROS appear as hypoxic signaling molecules. It seems paradoxical that a decrease in the level of the required substrate, O_2_, would result in an increase in ROS production. However, there is large evidence of its occurring [313,314,315].

Indeed, ROS appear to activate the hypoxia-inducible factor HIF-1α. Rho0 cells, which lack mtDNA and thus electron transport, do not show the HIF-1 DNA-binding activity following hypoxia [316]. Therefore, mitochondrial respiration is required for propagation of the hypoxic signal. Exogenous or endogenous H_2_O_2_ stabilize HIF-1α, thus inducing its activation during normoxia [316], suggesting that HIF stabilization and activation may be induced by ROS.

Many studies demonstrate activation of HIF-1α by ROS [317,318], that are supposed to act by inhibiting the prolyl-4-hydroxylase that addresses the factor to proteolytic digestion. Once stabilized, HIF-1α binds to hypoxia-responsive elements in the DNA and stimulates the expression of a large array of genes [319].

The oxidized cyt *c* level in the IMS seems to be essential to control mitochondrially-derived ROS [320,321]; accordingly, cyt. *c* in the IMS oxidizes superoxide produced by CIII to O_2_, thus preventing H_2_O_2_ production [321,322].

Inhibition of cytochrome *c* oxidase by hypoxia [323] enhances ROS formation by the respiratory chain; most likely because of the increased membrane potential [324] or by inducing a more reduced state of cyt. *c*.

The ROS produced by mitochondrial CIII seem to be critical for hypoxia signaling [294,325,326,327,328]. Hernansanz-Augustin et al. [329], however, showed that CI is involved in the superoxide burst under acute hypoxia in endothelial cells.

In addition, it should be pointed out that the plasma membrane NADPH oxidase [313,330] also seems to be involved in the enhanced ROS production during hypoxia.

### 5.4. ROS and Mitochondrial Quality Control

A major device for physiological control of the cells is the removal of damaged molecules; in particular, oxidized proteins are ubiquitinated and directed to the proteasome, a member of the ATP-dependent AAA+ proteases, where they are completely digested [331,332].

Mitochondrial proteins may particularly suffer from oxidative damage since they are close to the main sites of their generation, so that they must be removed in order to maintain mitochondrial integrity [333,334]. Internal mitochondrial proteins are translocated to the outer mitochondrial membrane, where they are ubiquitinated and then presented to the proteasome for degradation.

Mitochondrial dysfunction due to loss of cellular protein homeostasis (proteostasis) is a hallmark of aging and aging-related degeneration disorders [335], such as Alzheimer’s disease and Parkinson’s disease. Thus, mitochondrial proteins should be finely tuned. Although mitochondria form a proteasome-exclusive compartment [336], it is the cytosolic ubiquitin–proteasome system that plays a major role in the quality control of mitochondrial proteins.

The ATP-stimulated Lon protease in the mitochondrial matrix is devoted to the selective degradation of oxidized proteins, as for example the selective degradation of the Cox 4-1 subunit during hypoxia [337,338]. Defects in protein degradation have been involved in the age-related accumulation of oxidized proteins [339,340].

Also, entire cellular organelles undergo turnover and are finally directed to autophagy by digestion in the lysosome compartment [341,342,343].

Non-selective autophagy as well as selective mitophagy are triggered by ROS in response to several stressing signals [291,344,345]. Damaged mitochondria promote ROS generation, and excessive ROS can trigger mitophagy so as to remove impaired mitochondria and reduce ROS levels. Therefore, mitophagy helps to maintain cellular homeostasis under oxidative stress. Low ROS levels can trigger mitophagy in a mitochondrial fission-dependent way, if these levels are insufficient to trigger non-selective autophagy [346]. Therefore, a very specific and selective signaling cascade initiated by ROS has been suggested.

### 5.5. Supercomplexes Protect Complex I from ROS Damage and Limit ROS Generation

Our group first suggested that SC arrangement represents the missing link between oxidative stress and energy deficiency [96]. We speculated that oxidative stress dissociates the SCs, with loss of electron channeling and causing electron transfer to depend only upon the diffusion-coupled collisional encounters of the free ubiquinone molecules with the partner complexes.

#### 5.5.1. Supercomplex Association Protects from ROS Damage

As predicted by Lenaz and Genova [96], SC dissociation also induces disassembly of CI and CIII and consequent loss of their catalytic activity. Therefore, the alteration of electron transfer may stimulate ROS generation. Here we briefly summarize the experimental evidence supporting this hypothesis.

Analysis of the SCs in patients with an isolated deficiency of single complexes [347] and in cultured cell models harboring cytochrome *b* mutations [348,349,350] showed that SCs enhance the stability of CI. Genetic mutations causing loss of CIII prevent SC formation and determine a secondary loss of CI, and consequently primary CIII assembly deficiencies appear as joint CIII/I defects. On the other hand, however, the absence of CI does not affect CIII stability.

It was recently suggested that the absence of CIII blocks CI biogenesis by preventing the incorporation of the NADH module rather than decreasing its stability [351].

Most evidence, however, is in line with the idea that misassembly of CIII affects CI stability because of their physical interaction within the SC. Misassembled CIII would prevent SC formation, and a lack of SC would induce an enhanced ROS generation from CI (see Section 5.5.2), with consequent damage to CI itself [350,352] which is vulnerable to oxidative stress both directly and through the lipid peroxidation, particularly of CL [353]. According to Guaras et al. [120], saturation of CoQ oxidation capacity keeps a highly reduced state of CoQ itself and induces reverse electron transport from reduced CoQ to CI; the resulting local superoxide generation oxidizes specific CI proteins, triggering their degradation and the disintegration of the complex.

An ischemia/reperfusion injury enhances ROS generation in mitochondria which promotes the opening of the nonselective mPTP [354]. The mPTP opening further compromises cellular bioenergetics and increases mtROS, resulting in SC disintegration and ultimately leading to cell death.

A reconstitution study [71] showed that lipid peroxidation before formation of CI-CIII proteoliposomes abolishes the SC formation, as determined by flux control analysis. It is likely that the distortion of the lipid bilayer induced by peroxidation provokes the dissociation of the SC.

Mitochondrial ROS also affect CI and CIV activity through CL peroxidation in beef heart SMP [353,355]. CL liposomes exogenously added to mitochondria from aged rats almost completely restored the activity of these enzyme complexes to the values of young control animals [356,357]. Other phospholipid classes, as well as peroxidized CL, could not mimic the CL effect.

#### 5.5.2. Supercomplex Association Limits ROS Generation

Indirect considerations are compatible with the possibility that loss of SC organization may enhance ROS generation by the respiratory chain [12,96,358]. There are two possible reasons to support this hypothesis: a tight assembly of the respiratory complexes within SCs may screen auto-oxidizable groups hindering their reaction with oxygen (cf. [55]), and/or enhanced electron flow from NADH in the chain because of channeling would keep the prosthetic groups in a more oxidized form, thus preventing their interaction with oxygen [250]. On the other hand, on succinate oxidation, the reverse electron flow through CI keeps its centers more reduced favoring production of superoxide.

The molecular structure of the individual complexes and of the SCs does not provide structural evidence of such screening, since the prosthetic groups of CI in the matrix arm do not seem to be in close contact with CIII [359,360]. However, the SC I_1_III_2_IV_1_ from bovine heart has a different CI conformation with respect to the free complex, that appears to be bent in such way as to be closer to CIII [359]. This observation agrees with the notion that CI may undergo important conformational changes [361].

A different kind of indirect evidence on the protection exerted by SC organization against ROS generation comes from the observation that high mitochondrial membrane potential supports ROS generation, while uncoupling decreases ROS production [243,249]. Although these effects may have different explanations, they are compatible with the suggestion [87] that high membrane potential induces SC dissociation.

A direct demonstration that loss of SC organization enhances ROS production by CI was obtained in our laboratory [362] using a model system of reconstituted CI/CIII proteoliposomes at high lipid-to-protein ratio (30:1), where formation of the SC I_1_III_2_ is prevented. In this system, the generation of superoxide was much higher than in a similar system reconstituted at a 1:1 ratio, which is rich in SCs. In addition, we also dissociated the reconstituted proteoliposomes by mild detergent treatment using dodecyl maltoside and observed that the dissociation of CI was accompanied by a three/four-fold increase in ROS generation. An increase in ROS also occurs in mitochondrial membranes after detergent treatment.

A significant finding supporting our in vitro conclusions comes from a study of mitochondrial CI supramolecular structure in neurons and astrocytes. Lopez-Fabuel et al. [363] observed that CI is largely assembled into SCs in neuronal mitochondria, whereas astrocytes have higher content of free CI. The presence of free CI in astrocytes correlates with the severalfold higher ROS production by astrocytes compared with neurons. Thus, regulation of ROS generation by CI assembly into SCs may contribute to the bioenergetic differences between neurons and astrocytes. Nevertheless, ROS levels were found to be controlled by an efficient antioxidant system in astrocytes, to regulate redox signaling [364]. By exerting this control in ROS levels, metabolic functions are finely tuned in both neural cells.

A study of CI activity in the basidiomycete *Ustilago maydis* [365] suggested that the contacts between CI, CIII_2_ and CIV in the respirasome increase the catalytic efficiency of CI and regulate its activity to prevent ROS production.

The results of our proteoliposome study [362] are supported by many observations in cellular and animal models showing that SC dissociation is concomitant with enhanced ROS production.

Diaz et al. [352] showed enhanced ROS generation in mouse lung fibroblasts lacking the Rieske FeS protein of CIII and hence devoid of the SCs containing CI.

Mouse fibroblasts expressing the activated form of the *k*-ras oncogene and having low levels of high molecular weight SCs also produce higher ROS in comparison with wild type fibroblasts [366]. Hyperglycemia prevents SC formation, while increasing ROS levels in liver mitochondria of streptozotocin-diabetic rats. On the other hand, physical exercise was found to induce the chronic assembly of CIs into SCs in skeletal muscle [367], thus protecting mitochondria against oxidative damage.

Other studies in yeast mutants lacking the SC assembly factor Rcf1 and thus devoid of SCs CIII-CIV [117,368,369], showed enhanced ROS generation. Since the yeast *S. cerevisiae* lacks CI, in this case we may consider the origin of the extra ROS being presumably CIII. Thus, a major function of SC assembly appears to be this limitation of ROS production [370]: in the above case the decreased ROS generation by the Rcf1-stabilized SC III_2_-IV_2_ may be due to the more efficient electron transfer between CIII and CIV by cyt. *c*.

Several compounds including phospholipids, proteins, and certain chemicals are known to promote or stabilize mitochondrial SCs directly or indirectly [371], thus hindering ROS generation. Overexpression of the CL conversion enzyme ALCAT1 reduced SC formation and promoted ROS production, while preventing upregulation of coupled respiration [75]. These data suggest that the amount of ALCAT1 is critical for coupling mitochondrial respiration and metabolic plasticity.

The CL defect in Barth syndrome, a cardio-skeletal myopathy with neutropenia characterized by respiratory chain dysfunction, results in destabilization of the SCs, with higher levels of superoxide production in lymphoblasts from patients, compared to control cells [67,68]. Analogous results were obtained un studies on CL-lacking yeast mutants [372,373]. Moreover, Chen et al. [368] observed that yeast mutants, which cannot synthesize CL, exhibit increased protein carbonylation, an indicator of ROS.

Stein et al. [374] described a highly conserved 56-amino-acid microprotein named mitoregulin (Mtln). Mtln localizes to the mtIM where it binds CL and influences protein complex assembly. In cultured cells, Mtln overexpression increases mitochondrial SCs and mitochondrial activity while decreasing mitochondrial ROS.

The subunit composition of CI is crucial for SC formation and control of ROS production. Hou et al. [375] demonstrated that CI subunit AB1 (NDUFAB1), also known as mitochondrial acyl carrier protein, coordinates the assembly of respiratory CI, CII, and CIII, and SCs and is a crucial regulator of mitochondrial energy and ROS metabolism.

Jang and Javadov [376] observed that the pharmacological inhibition of CI and CII stimulated disruption of the respirasome accompanied by reduced ATP formation and increased ROS production. Overall, these studies provide biochemical evidence that the CI activity, and the NDUFA11 subunit are important for assembly and stability of the respirasome. The SDHC subunit of CII is not involved in the respirasome however the complex may play a regulatory role in respirasome formation.

Ramirez-Camacho et al. [377] found that mitochondria from reperfused hearts treated with *N*-acetyl-cysteine reduced oxidative stress and maintained SCs assemblies containing CI, CIII, CIV and the adapter protein SCAF1. The associations of the mitochondrial respiratory chain components into SCs could have pathophysiological relevance in metabolic diseases, as supramolecular arrangements, by sustaining a high electron transport rate, might prevent ROS generation [378].

## 6. Physiological and Pathological Implications

As we have pointed out in the Introduction, mitochondria are deeply involved in pathological processes. Mitochondrial dysfunction is often present in pathological states: at the end of this review, we intend to demonstrate that mitochondrial dysfunction is causative in the pathogenesis of most systemic diseases.

What are the signs showing mitochondrial dysfunction? There are many of them that are more or less specific. Among these we have changes of electron transfer, such as uncoupling, i.e., the occurrence of electron transfer not accompanied by ATP synthesis, changes of mitochondrial shape and number, alterations in quality control, changes in supramolecular structure (SCs) accompanied by decreased respiration and enhanced ROS generation, and finally opening of the mPTP and cell death.

We first briefly examine OXPHOS uncoupling. As for ROS generation cf. Section 5, for quality control see Section 5.4, and for the nature and role of the mPTP cf. Section 4.4. A possible role of SC disassembly as the initial pathogenetic event in many complex diseases will be described in Section 6.3.

### 6.1. Uncoupling: When Proton Movement and ATP Synthesis Are Disjointed

The term uncoupling in bioenergetics indicates that the H^+^ pumping activity of respiratory complexes which forms the Δ*p* is not exploited to synthesize ATP from ADP and Pi by F_1_F_O_-ATPase. Uncoupling is often associated with another term, mitochondrial dysfunction, which suggests that mitochondria do not work properly and are unable to fully perform their bioenergetic task. Uncoupling can have variable extent. We can distinguish between mitochondrial uncoupling, due to the downhill H^+^ flux through pathways outside the F_1_F_O_-ATPase, and intrinsic F-ATPase uncoupling, which means that the enzyme complex itself cannot match ATP synthesis to H^+^ channeling, due to molecular defects. While the mitochondrial uncoupling has a recognized physiological role, even if many mechanisms remain to be clarified [257], as far as we are aware the F_1_F_O_-ATPase uncoupling only represents a basic biochemical symptom of pathologies [379].

#### 6.1.1. Mitochondrial Uncoupling Due to Dissipative Pathways

The mitochondrial uncoupling can be defined as the detachment of H^+^ movement across the mtIM from the F_1_F_O_-ATPase activity [134]. Accordingly, the mitochondrial membrane potential generation is not used to build ATP. Even if it seems surprising, normally OXPHOS is not completely coupled, and the coupling extent depends on multiple factors. The mitochondrial uncoupling can be detected directly as a decrease in Δ*p* or indirectly as a decrease in the phosphorylation efficiency, namely in the ADP/O ratio and or in the respiratory control ratio (RCR), which corresponds to the ratios of State 3 (ADP-stimulated) and State 4 (basal) respiratory activities. State 4 respiration, after subtraction of the non-mitochondrial oxygen consumption detected upon respiratory chain inhibition, mirrors the oxygen consumption of the so-called proton leak. The latter consists in the futile cycle of H^+^ that from the IMS flow downhill in the mitochondrial matrix through pathways independent of the F_1_F_O_-ATPase [380]. The proton leak extent can be modified by protein complexes, exogenous compounds or permeability changes due to changes in the mtIM composition. Even if, at first, the mitochondrial uncoupling was only associated with mitochondrial dysfunction, at present, it emerges as key ruler of biological processes, as suggested by the occurrence of endogenous natural uncouplers, even if they can also have pathological meaning [257]. Accordingly, dissipative pathways normally occur in variable amounts in different cell types. In mammals, the most known physiological uncoupling occurs in the brown adipose tissue, where an integral membrane-bound protein (UCP-1) channels H^+^ and dissipates the electrochemical gradient resulting in thermogenesis. Other uncoupling proteins (UCP-2 and UCP-3) rule insulin secretion in pancreatic β cells and fatty acid metabolism in muscle, brown adipose tissue and heart, respectively [381]. Additionally, the four-membered family of adenine nucleotide translocators (ANTs), which catalyzes the ATP/ADP exchange across the mitochondrial membrane, possesses uncoupling properties [382].

Lastly, mitochondrial uncoupling can be also produced by exogenous chemicals, protonophores, namely lipophilic weak acids which can cross the mtIM, or non-protonophores, able to activate latent proton leaks.

A severe mitochondrial uncoupling leads to ATP depletion and eventually cell death, while mild mitochondrial uncoupling can have positive effects. Mitochondrial uncoupling can play opposite roles: it can promote cell death but also helps to protect cells against cell death, according to the cell type, the uncoupler and the uncoupling extent [257]. Uncoupling should prevent an excessive ΔΨ rise which would block electron transfer along the respiratory chain. The decrease in ΔΨ and consequently in Δ*p* also decreases ROS generation and vice versa, an increased ROS generation is known to decrease proton leak. On the other hand, the ATP synthase inhibition leads to electron accumulation in the respiratory complexes and ROS overproduction, which leads to oxidative stress and mitochondrial dysfunction. Endogenous mitochondrial uncoupling could prevent excessive ROS production. Accordingly, mitochondrial uncouplers have been tried to treat diseases featured by oxidative stress such as diabetes, obesity, cardiovascular diseases, neurodegenerative and aging-related diseases [134].

#### 6.1.2. Intrinsic ATP Synthase Uncoupling Due by Amino Acid Changes in the *a* Subunit

The term F-ATPsynthase/ase uncoupling refers to any condition that inhibits the coupling between the catalytic activity carried out by the hydrophilic portion and H^+^ translocation by the transmembrane portion F_O_ [383]. Frequently, it stems from structural changes that modify the H^+^ pathway of the F_1_F_O_-ATPase across the mtIM. The tight relationship between structure and function of mitochondrial complexes, described in the previous sections, means that any change in the primary sequence, due even to a mutation which changes a single amino acid can dramatically modify the protein properties, especially when amino acid substitutions involve crucial enzyme domains such as those in the *a* subunit where some amino acids are essential to build the H^+^ route. The mitochondrial F_1_F_O_-ATPase is a good example of how point mutations in some protein sectors result in severe diseases. Mutations in the nuclear genes that encode F_1_F_O_-ATPase subunits are rare and associated with severe diseases nearly incompatible with life. The most known mutations, associated with diseases whose severity is related to mitochondrial heteroplasmy [384], are localized in the mtDNA, which encodes *a* and A6L subunits of the F_O_ domain. In mammals, the mtDNA shows a higher mutational rate than nuclear DNA [385].

The most frequent mutations in the F_1_F_O_-ATPase associated with human pathologies occur in the mitochondrial *ATP6* gene, which encodes the *a* subunit. The structural arrangement of this subunit which, by positioning specific amino acids and exploiting the chemical properties of their side chains, forms the half-channels for H^+^ flow within the mtIM, remained enigmatic for years, thus making it difficult to envisage the link between altered molecular function and pathology. However, recent studies, which highlighted the H^+^ route, provided a satisfactory explanation on how point mutations are associated with mitochondrial dysfunctions and depicted a link between the bioenergetic defect and the syndrome. Up to now, several point mutations have been described [379]. The most severe mutation is the m.T8993>G transversion, namely the substitution of thymine by guanine, which results in the replacement of the hydrophobic leucine by the positively charged arginine, namely a missense mutation (*a*Leu156Arg) [386]. This molecular change is related to pathologies known as Neuropathy, Ataxia and Retinitis Pigmentosa (NARP) or Maternally Inherited Leigh Syndrome (MILS). These diseases are both associated with the same molecular defect, but exhibit various degrees of severity and are differently classified depending on the heteroplasmy degree [384]. The different chemical nature of the two amino acid side chains can satisfactorily explain the bioenergetic defect: since the inserted arginine is close to the crucial electrostatic barrier of *a*Arg-159, the two positive guanidine groups are close to each other to hamper both the H^+^ flux across the mtIM and ATP synthesis [140]. The consequence is a severe bioenergetic failure. Accordingly, the F_1_F_O_-ATPase becomes unable to pump H^+^ in the IMS and re-energize the mtIM, even if the two sectors F_1_ and F_O_ are still structurally and functionally joined, as proven by the observation that the enzyme complex remains sensitive to the selective inhibitor oligomycin, a clear evidence of the coupling of the two domains [387]. Similarly, the m.T9176>G transversion in the mitochondrial ATP6 gene that changes a conserved leucine into arginine (*a*Leu220Arg) on position 220 of *a* subunit [379] is associated with NARP and MILS diseases. In addition, in this case, as the *a*Leu-220 is close to the essential *a*Arg-159, this transversion changes the situation and makes two Arg residues occur in close positions, thus destabilizing the *a* subunit due to steric hindrance and electrostatic repulsions. Accordingly, the two vicinal Arg would act as a positively charged barrier, which prevents H^+^ translocation across the mtIM and decreases ATP synthesis and CIV respiration. Moreover, since ATP hydrolysis becomes uncoupled to H^+^ transport, as proven by the oligomycin insensitivity, the membrane potential cannot be restored by the F_1_F_O_-ATPase which cannot pump H^+^ [388]. Since these two transversions, which cause substitution of the hydrophobic side chain of Leu by a basic and positively charged chain of Arg, deeply alter the protein microenvironment and the H^+^ pathway, they cause the bioenergetic failure which constitute the biochemical basis of these severe diseases.

The m.T8993>C transition, which yields *a*Leu156Pro substitution [389], results in a less severe disease, as a result of an increased ROS production. In this case, the functionality is somehow preserved, since the *c*-ring can still slowly rotate, allowing the coupling of H^+^ flux to a low ATP synthesis [390]. It is most likely that the insertion of Pro which replaces Leu modifies the protein secondary structure. Accordingly, the Pro five-membered ring may cause a kink in the helices [391] which could slow down H^+^ transfer.

The de novo transition (m.G8969>A) in mtDNA which encodes the *ATP6* gene [392] has been recently associated with a rare mitochondriopathy, defined Myopathy, Lactic Acidosis, and Sideroblastic Anemia (MLASA) [392]. The consequent missense mutation Ser148Asn in *a* subunit [393] is localized at one helix turn from the *a*Glu-145 which acts as “H^+^ transfer group” in the half-channel which opens in the mitochondrial matrix [151]. The *a*Asn-145 bears a positive charge which makes ionic bond with *a*Glu-145, thus blocking H^+^ translocation which requires the –COOH deprotonation of *c*Glu-59 [393].

To sum up, the mutations in *a* subunit, which stepwise addresses H^+^ and allow H^+^ movement, block or hamper the torque generation in F_O_, which is essential for ATP synthesis by F_1_.

As far as we are aware, mutations in A6L subunit leading to pathologies are much less frequent than those in *a* subunit.

### 6.2. Supercomplexes and ROS Signaling

The role of mitochondrial ROS in cell signaling has been the subject of excellent reviews (cf. [272,274,394,395,396]). Here we deal with a possible role of the supramolecular organization of the respiratory chain on ROS signaling.

With ROS being involved in cell signaling, it is expected that their generation is subjected to tight control.

The control of mitochondrial ROS levels depends upon the balance between their rate of generation and of removal, as already considered in Section 5. The steady-state concentrations of the redox species responsible for electron leaking and ROS production are governed by a series of nuclear-encoded protein factors [267] and by the forces directly associated with respiratory activity, which are the redox potential of the NAD^+^/NADH couple and the Δ*p* [268]. Hoffman and Brookes [292] have investigated the ROS generation by rat liver mitochondria under different substrate and inhibitor conditions and different oxygen tensions, in order to determine the O_2_ affinity of the different O_2_-reacting sites: from such data, the apparent K_m_ for O_2_ was lowest for CI during forward flow, followed by CI backflow, CIII Q_O_ site, and highest for ETF dehydrogenase. They conclude that at physiological O_2_ concentration, only CI may be a significant source of ROS.

We first speculated [96] and then obtained experimental demonstration [71] that dissociation of SC I_1_III_2_ occurs upon ROS addition. Therefore, the facilitated electron channeling in the CoQ region is lost. This condition makes electron-transfer necessarily dependent upon random diffusion of the free ubiquinone molecules and collisions with the partner complexes and may elicit further ROS generation.

In fact, SC disorganization eventually leads to destabilization of CI, decreases NAD-linked respiration and ATP synthesis and increases superoxide production by CI (cf. also Section 5 for experimental evidence).

The study of Guaras et al. [120] pinpoints another aspect of SCs in relation to ROS formation. Hyper-reduction of the CoQ pool by ETFH_2_ oxidation during extensive fatty acid β-oxidation induces reverse electron transfer with a rise in ROS production by CI. Thus, shifting metabolic fuels from NADH-dependent to FADH_2_-dependent substrates may adjust ROS generation by way of the specific supramolecular assembly of the respiratory complexes involved [119,397].

### 6.3. Supercomplexes in Pathology and Aging

In order to understand the mechanism by which a mitochondrial dysfunction can lead to failure of physiological functions, it is paradigmatic to discuss the mechanisms responsible for aging of cells and organisms. The mitochondrial theory of aging [398,399,400] is based on a series of assumptions linked by a causal relationship [401]: (a) mitochondrial ROS generation is the major cause of aging. (b) Mitochondrial ROS damage mitochondrial biomolecules, especially inducing mtDNA somatic mutations, mainly affecting post-mitotic tissues. (c) The mtDNA mutations alter the structure of mtDNA-encoded proteins: since these are subunits of the major OXPHOS complexes, they imply a decreased oxidative phosphorylation, which leads to bioenergetic failure, metabolic derangement and cell death.

A further consequence of the decreased electron transfer during aging is further ROS generation, so that mtDNA damage and ROS production become involved in a vicious circle [398,399], not considered by the original theory.

Even if each of these assertions is sustained by experimental evidence, there are still several controversies and the cause–effect relationships among these events are still partially obscure. The complexity of biochemical, genetic, and regulatory systems and their inter-relationships are still not fully understood and often a single linear cause to effect relationship cannot be established [402]. Barja [403] however proposes that aging is the result of several effectors, i.e., mtROS production, lipid unsaturation, autophagy, mitochondrial DNA repair and putative other events such as apoptosis, proteostasis, or telomere shortening, already considered by different classic theories of aging. The aging regulating system gathers the theories of aging, previously considered as independent and assembles them into a single unified theory.

Wallace [1] hypothesized that mitochondrial dysfunction plays a central role in a wide range of age-related disorders and cancer types. He proposes that the delayed-onset and gradual onset of the age-related diseases is due to the accumulation of somatic mutations in the mtDNAs of post-mitotic tissues. The tissue-specific pathological phenotype may depend on the different roles and energy requirements of the various tissues. The individual varied predisposition to degenerative and cancer diseases may result from the interaction of present environment (dietary caloric intake) and ancient genetic factors (mitochondrial genetic polymorphisms). On these bases, the mitochondria make a direct connection between our environment and our genes.

Inherited and/or epigenomic variation of the mitochondrial genome determines our initial energetic capacity, but the age-related accumulation of somatic mtDNA mutations decreases the energetic capacity leading to disease [4], thus providing a unified pathophysiological and genetic mechanism for neurodegenerative diseases such as Alzheimer and Parkinson disease, metabolic diseases such as diabetes and obesity, autoimmune diseases, aging, and cancer.

Such an integrated model for the genetics and pathophysiology of complex diseases, aging, and cancer [4] is summarized in Figure 7.

Nuclear DNA variations including epigenetic changes, mtDNA variations, including ancient polymorphisms, and environmental influences including diet and calories, all impact the mitochondrial OXPHOS system. The primary defect is the reduction in energy conservation by OXPHOS, that in turn perturbs the mitochondrial biogenesis and enhances ROS production. ROS induce the progressive increase in mtDNA somatic mutations, so that the alteration of proteins in the OXPHOS complexes cause further decline of the mitochondrial function. If the mitochondrial energy production decreases below a specific tissue threshold, the bioenergetic failure can trigger cascade events which lead to cell death by apoptosis or necrosis. The clinical phenotype would result from the reduced energy storage in the tissues that require the most energy, such as brain, heart, muscle, and kidney. The symptom number and severity in these organs mirror the extent and type of the mitochondrial defect. The altered functions in mitochondria are also crucial to determine cancer initiation, promotion, and metastasis [404].

We first proposed SC disassembly as the missing link between oxidative stress and energy failure [96,405]. We proposed that an initial oxidative stress, not necessarily originating from mitochondria, and due to different possible reasons, causes dissociation of SCs. The bioenergetic consequence is the loss of electron channeling leading to random diffusion of ubiquinone, which is less efficient. A further consequence of SC disassembly would be CI dissociation with loss of electron transfer and/or proton translocation; the consequently altered electron transfer may result in further ROS generation. The possible severe metabolic and physiological consequences of SC dissociation are reported in the scheme in Figure 8 [223].

This model poses SC dissociation and ROS generation in a biunivocal situation. On the one hand, ROS cause dissociation of SCs, but on the other hand, SC dissociation enhances ROS generation. Clearly, these events must be tightly controlled, otherwise they might trigger a vicious circle of ROS generation. We have discussed above the hypothetic vicious circle of ROS generation and mitochondrial failure at the basis of aging and aging-related diseases. The original mitochondrial theory of aging did not imply the existence of a vicious circle [398], nevertheless the evidence that ROS generation increases in aging is overwhelming [403].

Alterations (dissociation) of the SCs often accompany pathological changes, however no direct proof exists that SC dissociation is the direct cause of the pathology.

Even so, circumstantial evidence points to a vicious circle (cf. Genova et al. [405] for review).

The major mitochondrial alterations accompanying aging and several pathologies occur at the level of CI [406,407], in line with the knowledge that this enzyme exerts the main control on respiration. Flux control analysis of respiration in coupled liver mitochondria [408] showed that the control exerted by CI is low in young rats, and becomes very high in the old ones, suggesting that aging induces CI structural changes that eventually affect the whole OXPHOS.

Analysis of the occurrence of respiratory SCs (cf. Section 3.1) in relation to age suggested that the SC destabilization plays a key role in the development of the aging-phenotype [409,410]. In mitochondria of rat cortex, a striking age-associated decline (−40%) in the CI-containing SCs, especially the SC I_1_III_2_ (−58%) was reported by Frenzel et al. [411].

The so-called mutator mice are a remarkable model of premature aging [407]; they have a defect of the proof-reading function of mtDNA polymerase-γ and hence exhibit multiple mtDNA mutations that result in decreased respiration and altered assembly of respiratory complexes [412]. In these mice, the steady-state level and activity of CI is strongly lowered, possibly as a secondary effect of a decreased assembly of CIV on the stability of CI, which disrupts the SC organization [349].

The observations collected in this review locate SC assembly/disassembly in a physiological signaling network that can easily undergo alterations, leading to dramatic conditions if ROS generation becomes out of control.

We may envisage SC association/dissociation as a physiological phenomenon, as depicted by the plasticity model [64], modulated by a variety of stimuli such as the mitochondrial membrane potential and protein post-translational changes in the respiratory complexes; thus, changes in ROS production modulate the signaling pathways started by ROS. These changes are reversible and must be tightly controlled.

We propose that the primary event responsible for aging and age-related pathologies is the structural damage induced by ROS in mitochondria, as originally stated by the so-called mitochondrial theory of aging [398].

Distinct events may constitute a consecutive series which leads to pathological changes. Progressive ROS-induced damage to the mitochondrial membrane lipids and proteins was found; mtDNA mutations, although present, may not necessarily be an early phenomenon of aging development.

The ROS level may be ruled by the nutrition state and the activity of the mTOR and insulin/IGF pathways. ROS may alter mitochondrial protein structure either directly or by means of peroxidation of CL, whose damage prevents SC association [71]; in turn SC dissociation would lead to further increase in ROS generation. If maintained at low levels, ROS may induce retrograde signals which promote the induction of compensatory mechanisms that try to counteract ROS generation and the consequent damage risk (see Section 5). However, high ROS levels may cause further damage and make the signaling pathways lose their coordination (see Section 5.3). At a later stage, mtDNA mutations would make the whole process irreversible and determine the final aging phenotype.

## Figures and Tables

**Figure 1 life-11-00242-f001:**
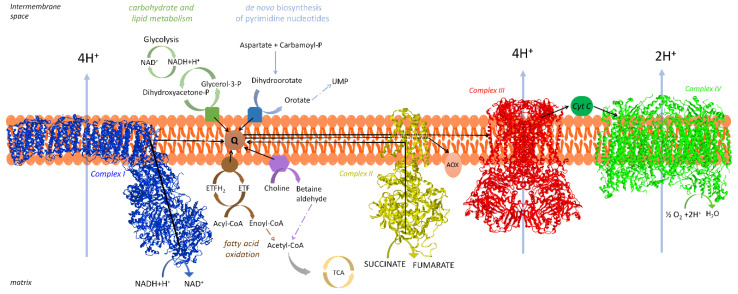
A schematic drawing of the respiratory chain depicting the protein complexes and their substrates. Complex I, Complex III and Complex IV are shown in their free form (modified PDB ID: 6YJ4, 2YBB, 1V54). Blue, CI, NADH-ubiquinone oxidoreductase; yellow, CII, succinate-ubiquinone oxidoreductase (modified PDB ID: 1ZOY); red, CIII, ubiquinol-cytochrome *c* oxidoreductase; green, CIV, cytochrome *c* oxidase; AOX, alternative oxidase; CoQ, Coenzyme Q (ubiquinone); Cyt*c*, cytochrome *c*. Four enzymes that reduce CoQ are also shown together with an indication of their metabolic pathways: from the intermembrane space (IMS), glycerol-3.P dehydrogenase (green) and dihydroorotate dehydrogenase (blue); from the matrix, electron transfer flavoprotein (ETF) dehydrogenase (brown) and choline dehydrogenase (purple). (see text for details).

**Figure 3 life-11-00242-f003:**
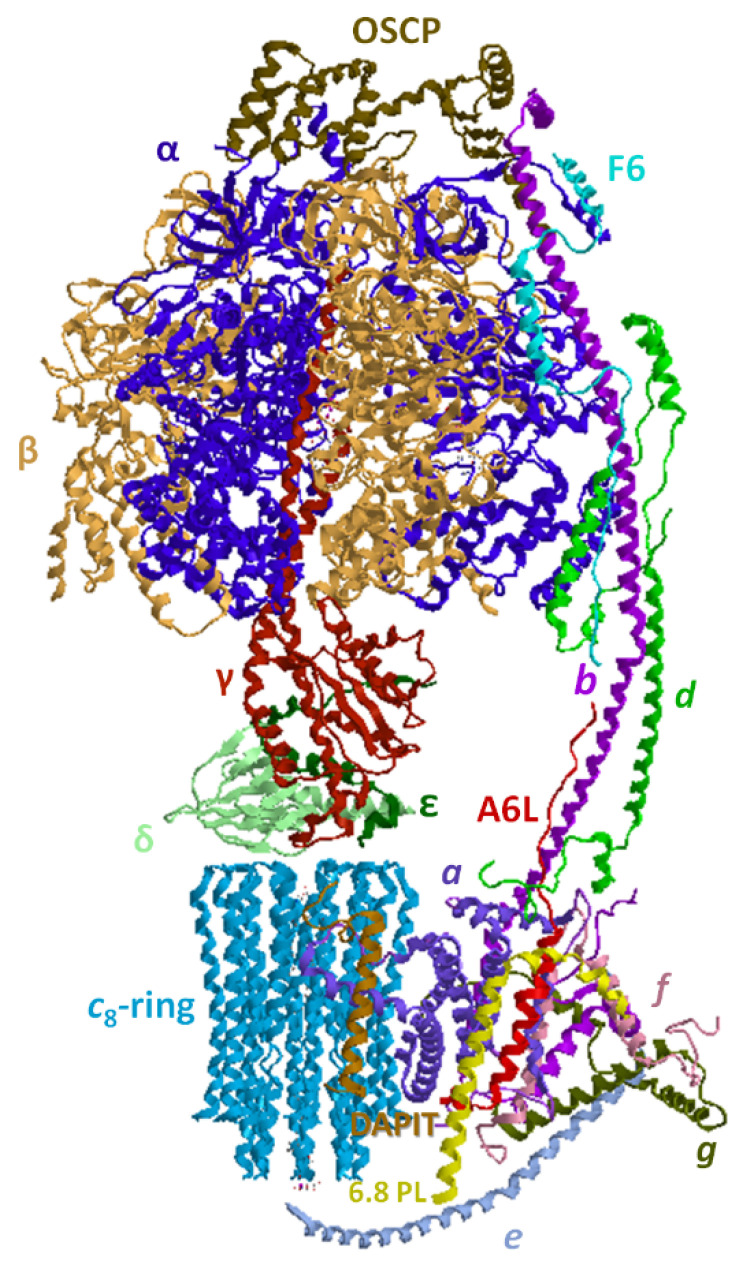
Structure of the mitochondrial F_1_F_O_-ATPase in mammals. The enzyme subunits are drawn as ribbon representations obtained from modified PDB ID code: 6TT7. The differently colored letters identify the subunits, drawn in the same color as the letter.

**Figure 4 life-11-00242-f004:**
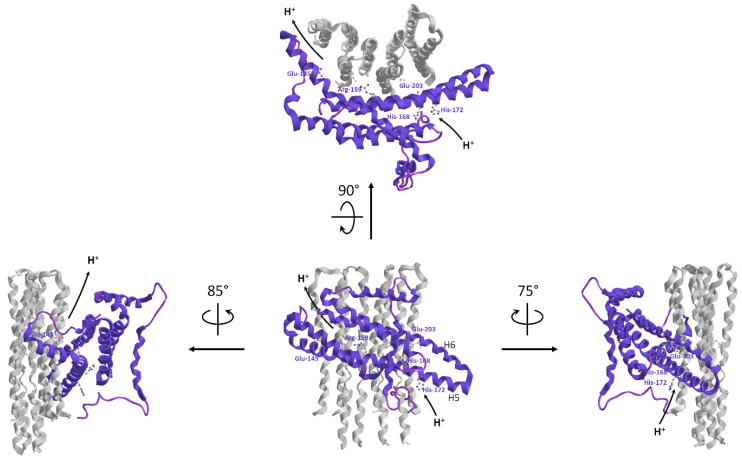
Proton translocation pathway within F_O_ during ATP synthesis. On the top the H^+^ entry and exit into and from the half channels, viewed from the IMS. On the left and right sides, the H^+^ outlet and inlet half-channels, respectively, after the rotation angles above the arrows are shown. The horizontal helices H5 and H6 of *a* subunit (violet) and four *c* subunits (gray), drawn as ribbon representations, were obtained from modified PDB ID code: 6TT7. The amino acids of *a* subunit involved in H^+^ translocation are drawn as ball and stick models.

**Figure 5 life-11-00242-f005:**
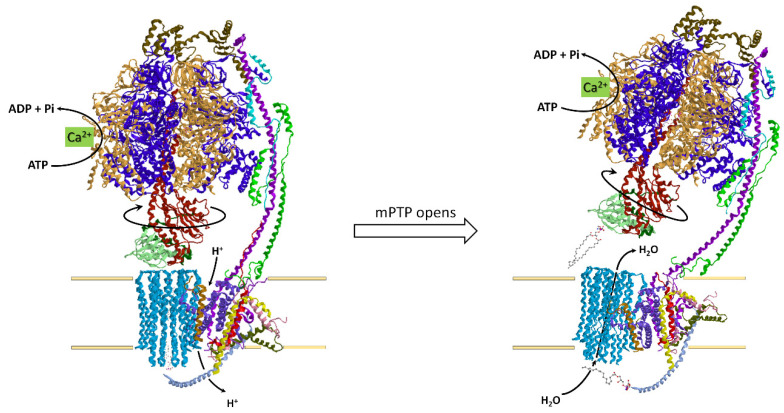
Model of mitochondrial permeability transition pore (mPTP) formation from the Ca^2+^-activated F_1_F_O_-ATPase. On the left Ca^2+^ bound to the catalytic sites activates the enzyme by triggering the structural change which opens the mPTP. On the right, the pore forms in the core of the *c*-ring when the lipid plug is pulled out. mPTP opening dissipates the mitochondrial Δ*p* and water entries in the matrix driven by oncotic pressure.

**Figure 6 life-11-00242-f006:**
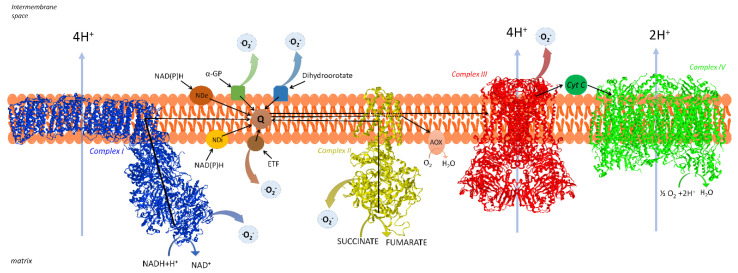
Major sites of superoxide production from the respiratory chain. The arrows represent the sources of superoxide at different sites in relation to the inner mitochondrial membrane. Blue, CI, NADH-ubiquinone oxidoreductase; yellow, CII, succinate-ubiquinone oxidoreductase; red, CIII, ubiquinol-cytochrome *c* oxidoreductase; green, CIV, cytochrome oxidase; NDi and NDe, internal and external alternative NAD(P)H dehydrogenases; AOX, alternative oxidase; αGP, glycerol-3-phosphate; ETF, electron transfer flavoprotein; Q, Coenzyme Q (ubiquinone); Cyt *C*, cytochrome *c*; (see text for details).

**Figure 7 life-11-00242-f007:**
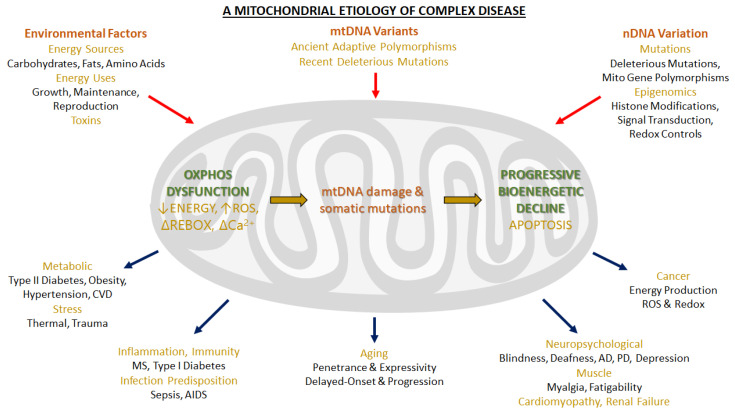
Integrated mitochondrial pattern which shows the genetic and phenotypic features of the “complex” human diseases. The model integrates the genetic and pathophysiological relationships of multi-factor diseases, aging, and cancer (from Wallace [4]). See text for explanations.

**Figure 8 life-11-00242-f008:**
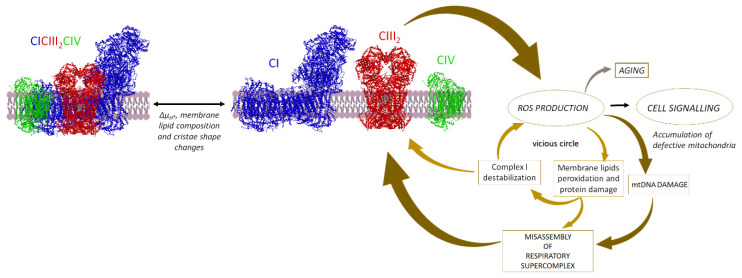
Scheme illustrating how the side chain (SC) organization loss may be involved in a vicious circle of oxidative stress and energy failure. Reactive oxygen species (ROS) production by CI increases due to SC disassembly. Other mitochondrial events, such as membrane phospholipid peroxidation, mtDNA damage and subsequent misassembly of the respiratory complexes with further loss of SC organization, may occur due to the increased oxidative stress, thus fueling and maintaining the vicious circle. In a dose-dependent way, ROS can also operate as molecular signals from mitochondria to the nucleus. See text for explanations.

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
