# Peer review of "Molecular and Supramolecular Structure of the Mitochondrial Oxidative Phosphorylation System: Implications for Pathology"

_life, 2021, doi:10.3390/life11030242_

Round 1

Reviewer 1 Report

This review describes in great depth the main structural and functional aspects of mitochondrial bioenergetics, discussing in particular, the recent progress on the supramolecular structure of the respiratory chain complexes and of F1FO-ATP synthase / hydrolase considering their physiological role and pathological implications.

I have no criticisms to make on the review that I consider excellent and in my opinion it is acceptable in its current version, I only recommend a quick review for few errors highlighted in the writing of some words, or words that are joined.

I just reported an error to correct on line 63 where the word "protons" is misspelled instead of the correct "electrons"

Author Response

We thank the reviewer for his positive and encouraging comment. We have tried to modify the mistake present in the text.

Reviewer 2 Report

The manuscript by S. Nesci et al. provides a comprehensive review of up-to-date literature on the mitochondrial energy transduction system encompassing structural aspects and their physiological and pathological relevance. The review is centered on the recent findings on the supramolecular organization of the oxidative phosphorylation system and is divided into different parts, namely respiratory chain supercomplexes, the ATP synthase and its supramolecular organization, the mitochondrial production of reactive oxygen species and finally the structural and functional alterations that give rise to pathological changes. Special emphasis is given to the hypothesis that changes in the supramolecular organization are linked to decreased energy yield and increased generation of damaging species, thus causing aging and age-related pathological changes.

This is an informative, well-written and very timely review with interesting historical detours. It would be appealing to a wide audience of cell biologists and especially useful to young researches. I highly recommend its publication provided that the following minor  comments are dealt with properly.

The manuscript follows a logical pattern, but there is some lack of coordination between the different parts. It is suggested to explicitly state the logical links existing between them. For example, it should be stated how the supramolecular structure of the ATP synthase is linked to the pathological changes described in the last section.

The authors describe the supramolecular arrangement of the ATP synthase defining the structural organization of its subunits (para. 4.3), but miss to explain the defects in the shape of the inner mitochondrial membrane that result in severe human diseases. 

The possible interaction of some MICOS components with the ATP synthase in the process of cristae biogenesis should be briefly mentioned.

The article would benefit from English language editing as there are some awkward phrases and typos.

Author Response

According to the precious suggestions of the reviewers, the text has been carefully checked and all typos, mispelling and mistakes which were found were corrected. All changes are in red. Some tortuous sentences have been split into two sentences to achieve clarity. Some colloquial phrases were deleted or replaced. Abbreviations and acronyms are now more extensively used and have a more homogenous aspect. The spelling of some words which can be written in two ways has been uniformed to avoid confusion. Some introductory and connecting sentences have been added to the text to make the link among the various sections more clearly emerge and improve their coordination, as suggested by Reviewer 2. Accordingly, in the revised version the involvement of the supramolecular structure in the mitochondrial shape and in pathological changes is now outlined. This is an emerging topic and unfortunately literature data are scanty.

The possible interaction of some MICOS components with the F1FO-ATPase has been briefly described.

We heartily hope that now our work is significantly improved.
